# Representation Alignment in Neural Networks

**Ehsan Imani**                                                              *imani@ualberta.ca*
*University of Alberta*

**Wei Hu**                                                                        *vvh@umich.edu*
*University of Michigan*

**Martha White**                                                          *whitem@ualberta.ca*
*University of Alberta*
*CIFAR AI Chair*

**Reviewed on OpenReview:** *https://openreview.net/forum?id=fLIWMnZ9ij*

## Abstract

It is now a standard for neural network representations to be trained on large, publicly available datasets, and used for new problems. The reasons for why neural network representations have been so successful for transfer, however, are still not fully understood. In this paper we show that, after training, neural network representations align their top singular vectors to the targets. We investigate this representation alignment phenomenon in a variety of neural network architectures and find that (a) alignment emerges across a variety of different architectures and optimizers, with more alignment arising from depth (b) alignment increases for layers closer to the output and (c) existing high-performance deep CNNs exhibit high levels of alignment. We then highlight why alignment between the top singular vectors and the targets can speed up learning and show in a classic synthetic transfer problem that representation alignment correlates with positive and negative transfer to similar and dissimilar tasks. A demo is available at `https://github.com/EhsanEI/rep-align-demo`.

## 1 Introduction

A common strategy for transfer learning is to first learn a neural network on a source (upstream) task with a large amount of data, then extract features from an intermediate layer of that network and finally train a subsequent model on a related target (downstream) task using those extracted features. The premise is that neural networks adapt their intermediate representations—hidden representations—to the source task and, due to the commonalities between the two tasks, these learned representations help training on the target task (Bengio et al., 2013). Availability of large datasets like ImageNet (Russakovsky et al., 2015) and the News Dataset for Word2Vec (Mikolov et al., 2013) provides suitable source tasks that facilitate using neural networks for feature construction for Computer Vision and Natural Language Processing (NLP) tasks (Kornblith et al., 2019; Oquab et al., 2014; Devlin et al., 2018; Pennington et al., 2014).

There is as yet much more to understand about when and why transfer is successful. Understanding the properties of the learned hidden representations and their benefits for training on similar tasks has remained a longstanding challenge (Touretzky & Pomerleau, 1989; Zhou et al., 2015; Marcus, 2018). One strategy has been to define properties of a good representation, and try to either measure or enforce those properties. Disentanglement and invariance are two such properties (Bengio et al., 2013), where the idea is that disentangling the factors that explain the data and are invariant to most local changes of the input results in representations that generalize and transfer well. Though encoding properties for transfer is beneficial, it remains an important question exactly how to evaluate the representations that do emerge.

One challenge is that even hidden representations of two neural networks trained on identical tasks appear completely different, and studying the representations requires measures that separate recurring properties from irrelevant artifacts (Morcos et al., 2018). One direction has been to analyze what abstractions the network has learned, agnostic to exactly how it is represented. Shwartz-Ziv & Tishby (2017) studied neural networks through the lens of information theory and found that, during training, the network preserves the information necessary for predicting the output while throwing away unnecessary information successively in its intermediate layers. Using representational similarity matrices, Hermann & Lampinen (2020) found that on synthetic datasets where task-relevance of features can be controlled, learned hidden representations suppress task-irrelevant features and enhance task-relevant features. Neyshabur et al. (2020) showed that neural networks trained from pre-trained weights stay in the same basin in the loss landscape. In reinforcement learning, Zahavy et al. (2016) explained the success of Deep Q-Networks by visualizing how the learned hidden representations break down the input space in a way that respects the temporal structure of the task. Analyses of NLP models have found linguistic information in the hidden representations after training (Belinkov et al., 2017; Shi et al., 2016; Adi et al., 2017; Qian et al., 2016).

Other works have focused on individual features in the learned representations. Saliency maps and Layer-Wise Relevance Propagation (Simonyan et al., 2014; Zeiler & Fergus, 2014; Bach et al., 2015) that show the sensitivity of the prediction to each unit in the model are popular in Computer Vision and demonstrate the appearance of useful features like edge or face detectors in neural network. In NLP, Dalvi et al. (2019) studied the relevance of each unit to an external task or the model's own prediction.

In this work we look at singular value decomposition of the learned features and find that **after training, neural network hidden representations align their top singular vectors to the task**. We discuss this observation in detail in Section 4. Section 5 provides an empirical study using different optimizers, architectures, datasets, and other design choices to support this claim. There are a number of previous theoretical results on optimization and generalization that exploit alignment between the top singular vectors and the task (Arora et al., 2019; Oymak et al., 2019a; Cortes et al., 2012; Canatar et al., 2021). While these results were originally motivated for neural tangent kernels and other kernels, our observation shows that hidden representations of a neural network also satisfy the conditions for these results. In section 6, we discuss one of these previous results, a convergence rate developed by Arora et al. (2019) and Oymak et al. (2019a), and describe how our observation makes it applicable for studying feature transfer. Finally, in Section 7 we study feature transfer first in a controlled synthetic benchmark and then in pre-trained CNNs and find that positive and negative transfer can be traced to an increase or decrease in alignment between the learned representations and the target task.

## 2    Problem Formulation and Notation

We focus on the transfer setting. We assume there is a source task where data is abundant and a target task with a small number of available data points. A neural network is learned on the source task, and the final hidden layer used as learned features. These learned features are then given as the representation for the target task, on which we learn a linear model. More generally, a nonlinear function—like another, likely simpler, neural network—could be learned, but we focus on linear models.

More formally, we consider a dataset with $n$ samples and input matrix $X_{n \times m}$ and label vector $y_{n \times 1}$. We assume we have a (learned) feature function $\phi : \mathbb{R}^m \to \mathbb{R}^d$ that maps inputs to features, with representation matrix $\Phi_{n \times d}$ where $n \geq d$. Without loss of generality, we replace the bias unit with a constant feature in the matrix to avoid studying the unit separately. We learn a linear model, with weights denoted by $w_{d \times 1}$, learned using the squared error in regression and logistic loss in classification. In regression, $y \in \mathbb{R}^n$ and the model $w$ is learned using $\|\Phi w - y\|^2$, where $\|\cdot\|$ is the $\ell_2$ norm. In classification, $y \in \{-1, +1\}^n$ and the logistic loss is $\sum_{i=1}^{n} \log(1 + \exp(-\phi_i^\top w \cdot y_i))/\log(2)$.

Alignment will be defined in terms of the singular vectors of $\Phi$ and label vector $y$. The singular value decomposition (SVD) of $\Phi$ is $U\Sigma V^\top$, where $\Sigma_{n \times d}$ is a rectangular diagonal matrix whose main diagonal consists of singular values $\sigma_{1:d}$ that are in descending order and $U_{n \times n}$ and $V_{d \times d}$ are orthonormal matrices whose columns $u_{1:n}$ and $v_{1:d}$ are the corresponding left and right singular vectors. Implicitly, the singular values $\sigma_{d+1:n}$ are zero, giving this decomposition where $\Sigma$ is rectangular. For a vector $a$ and orthonormal

basis $B$, $a^B$ denotes $B^\top a$ or the representation of $a$ in the basis $B$. We will use $r(\cdot)$ to denote the rank of a matrix.

## 3 Beyond Expressivity

A set of expressive representations may differ in terms of optimization behavior when learning a linear function on those features. In particular, some representations may improve the convergence rate, requiring fewer updates to learn the linear function. There are also theoretical results that relate this improved convergence rate with improvement in generalization ability (Hardt et al., 2016).

Let us clarify this point with an experiment before moving to the transfer learning setting. We sampled 1000 points from the UCI CT Position Dataset (Graf et al., 2011) and created three sets of 1024-dimensional features. The first representation is extracted from a neural network with one ReLU hidden layer. The second one is obtained with sparse dictionary learning (Mairal et al., 2009). The third one is the representation learned by a two-layer RBF network. All of these representations are nonlinear and high dimensional and a linear model on each one can perfectly fit this data. We verified this by evaluating the closed form least squares solution on the extracted representations.

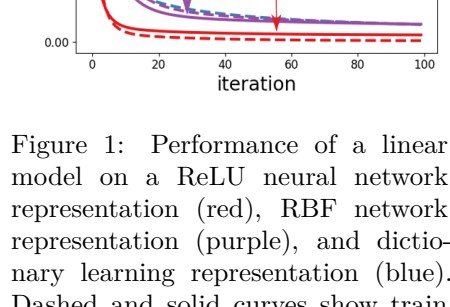

Figure 1: Performance of a linear model on a ReLU neural network representation (red), RBF network representation (purple), and dictionary learning representation (blue). Dashed and solid curves show train and test mean squared errors. We swept over four different step-size values in $\{0.01, 0.1, 1, 10\}$ and chose the best performing model.

However, as shown in Figure 1, training a zero initialized linear model with gradient descent on these representations shows interesting differences between them. First, optimization is faster when the representation is extracted from a neural network. The second difference is in the error of the models trained on these representation when they are evaluated on a separate test set of 1000 points. These differences—as opposed to expressivity, linear separability, or the ability to fit the data—are the subject of this study.

In the next section we will introduce a measure called representation alignment to evaluate the quality of a representation in terms of the convergence rate and generalization ability that it provides. Neural networks increase alignment in their hidden representation during the training, more so in hidden layers closer to output. We will then discuss how training with gradient descent on representations with high alignment (such as neural network representations) results in an initial fast phase that reduces most of the loss with small weights.

## 4 Representation Alignment

Representation alignment, as defined in this work, is a relationship between a label vector and a representation matrix. Representation alignment facilitates learning the model $w$. In this section, we define and discuss alignment, starting first by providing intuition about the reason for the definition.

To understand alignment, consider first the squared loss, rewritten using the SVD

$$\left\|\Phi w - y\right\|^2 = \left\|U\Sigma V^\top w - UU^\top y\right\|^2 = \left\|U\left(\Sigma w^V - y^U\right)\right\|^2 = \left\|\Sigma w^V - y^U\right\|^2$$

$$= \sum_{i=1}^{r(\Phi)} \left\|\sigma_i w_i^V - y_i^U\right\|^2 + \sum_{i=r(\Phi)+1}^{n} \left\|y_i^U\right\|^2 \tag{1}$$

The vector $y^U$ is the projection of the entire label vector $y$ of $n$ samples onto the basis $U$. In this rewritten form, it is clear that if $\sigma_i$ is small and the rotated $y_i^U = u_i^\top y$ is big, then $w_i^V$ has to become very big in order to reduce the loss in this task, which is problematic. The representation $\Phi$ is aligned with the label vector $y$ if $u_i^\top y$ is large primarily for the large singular values $\sigma_i$.

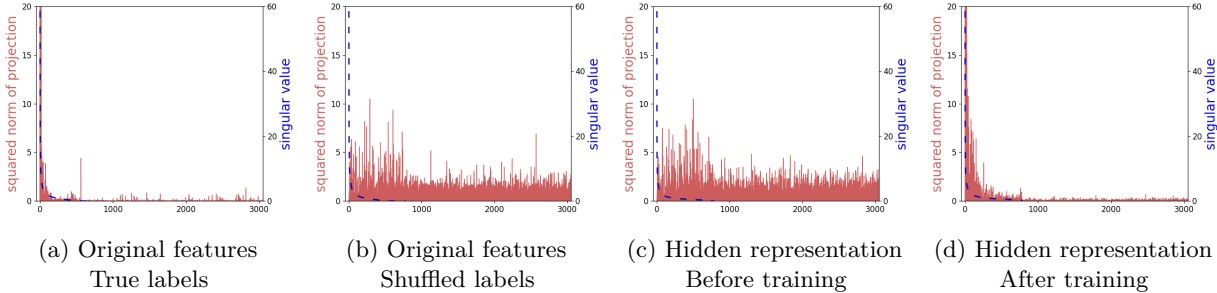

(a) Original features
True labels

(b) Original features
Shuffled labels

(c) Hidden representation
Before training

(d) Hidden representation
After training

Figure 2: Projection of the label vector on different directions. The horizontal axis shows the index of the singular value and is zoomed in to the region of interest. The representations are normalized to length one for proper comparison. Comparing a and b shows the alignment in MNIST that is broken when the labels are shuffled. For c and d, the network is trained on shuffled labels, and comparing them shows emergence of alignment between the hidden representation and the training labels.

Before giving a formal definition, let us visualize these values. In Figure 2 (a) we sampled 10000 points from the first two classes of the MNIST dataset (5000 points from each class) and plotted the singular values of $X$, the original features in the dataset, and the squared dot product between the label vector $y$ and the corresponding left singular values $u_{1:n}$. The dot product is noticeably large for the top few singular vectors, and drops once the singular values become small. This need not always be the case. We created the same plot for shuffled labels in Figure 2b. As the association between the features and the labels in MNIST dataset is lost, the label vector is more or less uniformly aligned with all the singular vectors.

We define alignment relative to a threshold on singular values. We assume $\|x_i\| \leq 1$ for all $i \in \{1, \ldots, n\}$, which can be satisfied by normalization. This constraint ensures singular values are not large simply because feature magnitudes are large, and so facilitates comparing different representations. Given a threshold $\tau \geq 0$, the degree of alignment between the representation $\Phi$ and label vector $y$ is

$$\text{Alignment}(\Phi, y, \tau) \doteq \sum_{\{i:\sigma_i \geq \tau\}} (u_i^\top y)^2 \tag{2}$$

If $\text{Alignment}(\Phi, y, \tau)$ is larger for higher $\tau$, then the projected label vector is concentrated on the largest singular values.

We visualize hypothetical good and bad alignment in Figure 3, if we were to plot alignment versus the threshold. The green line depicts a hypothetical curve that has higher alignment, and the red bad alignment. The curves start at the same point because at a threshold of 0 all singular values are greater than or equal to the threshold, so the sum in equation 2 includes the sum over all terms $(u_i^\top y)^2$. As the threshold increases, alignment quickly drops for the red curve, indicating that the projected label vector is not concentrated on the largest singular values. Rather, it is spread out across many small singular values, which we later motivate results in slow learning. In the green curve, even for a relatively high threshold, the sum still includes most of the magnitude, indicating $(u_i^\top y)^2$ is small for small singular values.

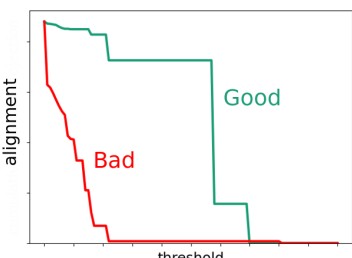

Figure 3: A hypothetical alignment-threshold curve.

When comparing alignment curves in our later experiments, we often see a clear difference between the two curves across a wide range of thresholds as in Figure 3. In these cases, for brevity, we simply state that the green curve has higher alignment without referencing the thresholds.

## 4.1 An Illustrative Example

Alignment in high-dimensional representations can be difficult to visualize. Before measuring alignment in neural network representations, it can be helpful to understand when alignment may be low or high. Figure 4

is an example that illustrates how the relationship between the representation and the label vector affects alignment. A set of 1000 points are normalized to unit length so they fall on a circle. 500 points have angles around $\pi/4$ and the angles for the other 500 points are around $-3\pi/4$. This representation matrix has two right singular vectors (principal components) $v_1$ and $v_2$ with singular values $\sigma_1 = 9.58$ and $\sigma_2 = 2.84$. The left and right plots shows two different possible sets of labels, where yellow is for class one and purple for class two.

The labeling on the left defines the two classes in the direction of the first principal component, where data is more spread out. As a result, the label vector is mostly in the direction of the first singular vector $u_1$. The labeling on the right defines the classes in the direction of $v_2$ so the classes are closer together. The resulting label vector is mostly in the direction of $u_2$ which has a smaller singular value. Both representations are perfectly able to linearly separate the two classes, but the alignment is higher in the leftmost plot for any threshold between $\sigma_1$ and $\sigma_2$.

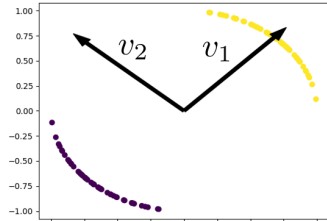 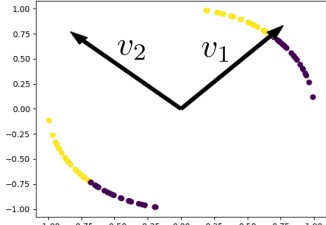

Figure 4: The labeling that separates the two classes in the directions where data is more spread out (left) is mostly in the direction of the first singular vector with singular value $\sigma_1 = 9.58$. Recall that the weights for the separating hyperplane are orthogonal to the plane; here $v_1$ is orthogonal to the plane that separates these two classes. The labeling on the right is mostly in the direction of the second singular vector with singular value $\sigma_2 = 2.84$. It has to use the direction with smaller singular value to fully separate the two classes.

## 4.2 Emergence of Alignment in Learned Features

Now let us return to alignment in the representations learned by neural networks. In our first example, in Figures 2 (a) and (b), we used the raw inputs as the features and measured their alignment to the label vector. We could ask if neural network representations improve on this alignment, but because the alignment is already very good, we are unlikely to see notable improvements. Instead, we can synthetically create a much more difficult alignment problem by shuffling the labels. We can ask if alignment can emerge using neural networks even in the absence of good alignment between the input matrix and label vector, i.e., when the labels are shuffled.

We trained a neural network with three hidden layers as wide as the input layer using Adam (Kingma & Ba, 2014) for 1000 epochs with batches of size 64 on the shuffled labels and extracted the representation from the final hidden layer. Figures 2 (c) and (d) show the squared projection of the training label vector on the singular vectors of the hidden representation matrix before and after training.

The network has learned a representation where the label vector is mostly aligned with the top singular vectors. Recall that this is for shuffled labels. This comparison illustrates emergence of representation alignment in hidden representations, even when there is no such relationship between the label and the input features.

One might wonder if this is a property of all representation learning approaches, or particular to neural networks. We have already seen that sparse coding with dictionary learning does not appear to have

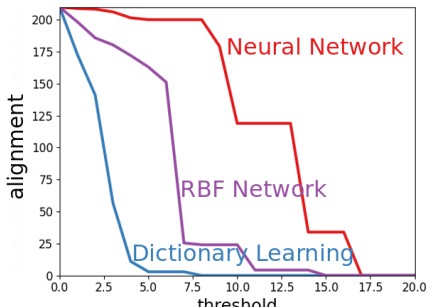

Figure 5: Alignment of the ReLU NN (red), RBF network (purple), and sparse coding (blue) representations.

this property, and in fact when we measure the alignment for Figure 1, we find it is much lower than that of the neural network. But sparse coding is an unsupervised procedure. Perhaps whenever we include a supervised component, alignment improves.

The learned RBF network allows us to assess this hypothesis, since it is supervised but has a different architecture than typical neural networks. In particular, its first layer consists of RBF units and training adjusts the centers of these RBF units. This RBF network could similarly improve alignment, through training, and provides a comparison point for whether current NN architectures are particularly able to improve alignment.

We find that alignment for the RBF network is better than sparse coding, but still notably worse than the NN. All the curves start at 200, because at a threshold of 0 all singular values are greater than or equal to the threshold, so the sum in equation 2 includes the sum over all terms $(u_i^\top y)^2$. As the threshold increases, alignment quickly drops for dictionary learning, indicating that the projected label vector is not concentrated on the largest singular values.

Rather, it is spread out across many small singular values, which we later motivate results in slow learning. The RBF network does not drop as quickly, but its curve is more similar to dictionary learning than the NN. The NN has notably better alignment. At a threshold of about 8, the sum still includes most of the magnitude, indicating $(u_i^\top y)^2$ is small for singular values smaller than 8.

This one experiment is by no means definitive. We include it to show that different feature learning methods can have different alignment properties. This work is primarily focused on showing the emergent alignment properties of neural networks, rather than making the much stronger claim that this is a unique property to neural networks. But it is motivating that, even in this simple setting, an NN has significantly better alignment; it is a phenomenon worth investigating.

## 5    Experiments Measuring Alignment in a Variety of Networks

We provide an empirical study of representation alignment with various architectures and optimizers for regression in the UCI CT Position Dataset and for classifying the first two classes of Cifar10 and MNIST. We then investigate the degree of alignment in internal layers of the learned networks.

**Training increases alignment in different architectures and optimizers.**    This section shows one of our main results: neural networks, across depths, widths, activations and training approaches, significantly increase alignment. We test five different settings. In the first, we fix the hidden layer width to 128, the optimizer to Adam and train networks of different depth. In the second, we set the depth to 4, the optimizer to Adam and train networks of different hidden layer width. In the third, we set the depth to 4, the hidden layer width to 128 and train networks with different optimizers. The fourth and fifth settings consider other activations (tanh, PReLU, LeakyReLU, and linear) and batch-sizes (32, 128, and 256). In all of these settings, training increases alignment in the final hidden representation compared to both the input features and the hidden representation at initialization. The increase in alignment occurs on both train and test data. Interstingly, the increase in alignment happens even with linear activation where the successive layers cannot make the data more binary separable. The plots for train data across different architectures and activations are shown in Figure 6 and the rest of the plots are in Appendix A. Note that the networks in this figure do not have convolutional layers, residual connections or batch normalization. Adding depth to these architectures might have no positive effect or even harm performance (Ioffe & Szegedy, 2015; He et al., 2016), which could be why we do not find a consistent increase in final alignment as depth increases.

**Alignment is higher in layers closer to the output.**    Conventional wisdom in deep learning is that networks learn better representations in the layers closer to output (Goodfellow et al., 2016). Alain & Bengio (2017) trained linear classifiers on the learned hidden representations and showed that linear separability increases along the depth. Another example is in Computer Vision where the first hidden layers of a CNN usually find generic features like edge detectors, and the last hidden layer provides features that are more specialized and allow the final layer to achieve high performance (Krizhevsky et al., 2009; Yosinski et al., 2014).

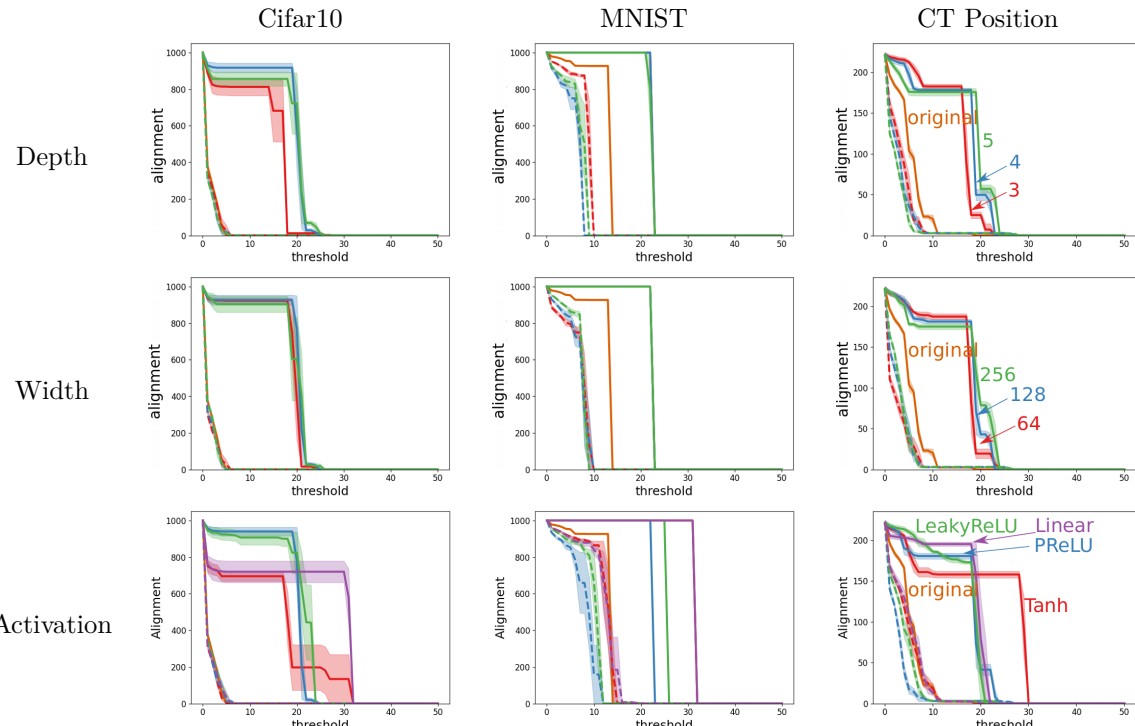

Figure 6: Training increases alignment in hidden representation regardless of architecture. The orange line is the input features, the dashed lines are hidden representations before training, and the solid lines are hidden representations after training. The plots are averaged over 5 runs and the shaded area shows standard errors. The label vector for the regression task was moved to the interval $[-1, 1]$ for easier comparison with classification. The curves for learned representation on MNIST mostly overlap as all models can achieve a high level of alignment on this task.

We test the hypothesis that training a multi-layer network results in hidden representations that are successively better aligned with the labels. For MNIST and CT Position, we pick 10000 random points and train a neural network with three hidden layers of width 128 using Adam with batch-size 64 until convergence. Figure 7 shows the alignment for all the hidden representations along with the original features before and after training, averaged over 5 runs.

On both datasets, layers closer to the output are monotonically better aligned after training. Comparison with the plots at initialization shows that this pattern is the result of training on the task. In fact, the ordering is completely reversed before training. This is probably due to the loss of information after multiple layers of random transformation in the initialized network. We will also discuss this in a later section.

# 6 Alignment and Convergence Rates

In this section we discuss how aligned representations help speed up learning. We first conceptually explain this phenomenon, by reviewing a fine-grained convergence rate. We then provide a simple experiment to highlight the connection between alignment and convergence rates.

## 6.1 Improving Convergence Rates

The role of alignment in optimization and generalization has been recently noted (Jacot et al., 2018; Arora et al., 2019; Oymak et al., 2019a; Canatar et al., 2021) in overparameterized networks for the standard training-testing setting. We leverage these insights for the transfer setting, where a neural network—that need not be overparameterized—is learned to create features for transfer, to a new target task. The aim

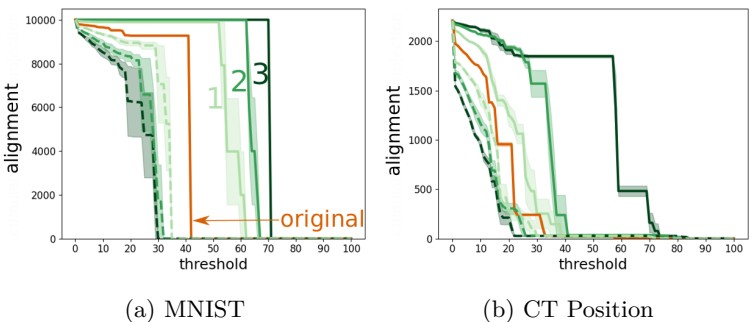

(a) MNIST     (b) CT Position

Figure 7: Alignment of input and hidden layers before (dashed) and after (solid) training on MNIST and CT Position. Training increases alignment more so for layers closer to the output.

of this paper is to show emergence of alignment in neural network hidden representations and characterize positive and negative transfer in this framework.

To understand how alignment can help with convergence rates in the optimization, let us review the following fine-grained convergence rate.

**Proposition 1.** *Given full rank representation matrix $\Phi$ and label vector $y$, let $w^* = (\Phi^\top \Phi)^{-1} \Phi^\top y$ be the optimal linear regression solution and $\sigma_{max}$ be the maximum singular value for $\Phi$. The batch gradient descent updates $w_{t+1} = w_t - \eta \Phi^\top (\Phi w_t - y)$ with stepsize $0 < \eta < \sigma_{max}^{-2}$ and $w_0 = 0$ results in*

$$\|\Phi w_t - \Phi w^*\| = \sqrt{\sum_{j=1}^{r(\Phi)} (1 - \eta \sigma_i^2)^{2t} (u_i^\top y)^2} \tag{3}$$

$$\|\Phi w_t - y\|^2 = \sum_{j=1}^{r(\Phi)} (1 - \eta \sigma_i^2)^{2t} (u_i^\top y)^2 + \|\Phi w^* - y\|^2$$

Arora et al. (2019) showed this result for an input Gram matrix, similar to $\Phi \Phi^\top$, and that the convergence rate is exactly dictated by alignment. Their analysis was for infinitely wide ReLU networks. Our goal is not to measure alignment within the network itself, but rather of the representations produced by the neural network. We therefore provide the above proposition for features $\Phi$ given by the neural network; the proof is straightforward and given in Appendix B.

The following proposition uses the result above and relates the convergence rate to representation alignment as defined in this paper and highlights the role of threshold. The use of 0.9 in the statement of this proposition is to avoid excessive notation in the main paper and the proof in Appendix B holds for a generic ratio by including a logarithmic factor.

**Proposition 2.** *Under the conditions of the previous proposition, if $Alignment(\Phi, y, \tau) = \delta$ for a threshold $0 < \tau \leq \sigma_{max}$, then gradient descent needs at most $O(1/(\eta \tau^2))$ iterations to reduce the loss by $0.9\delta$.*

Let us examine why Proposition 1 highlights the role of alignment in convergence rates. Notice that if $(u_i^\top y)^2$ is large for a larger singular value $\sigma_i$, then $(1 - \eta \sigma_i^2)^{2t}$ is smaller and makes the product in the sum in Equation 4 smaller. Having a smaller sum means that $w_t$ is closer to $w^*$ after $t$ iterations. On the other hand, if $(u_i^\top y)^2$ is large for a small singular value $\sigma_i$, then it is multiplied by a larger number $(1 - \eta \sigma_i^2)^{2t}$, making this sum larger.

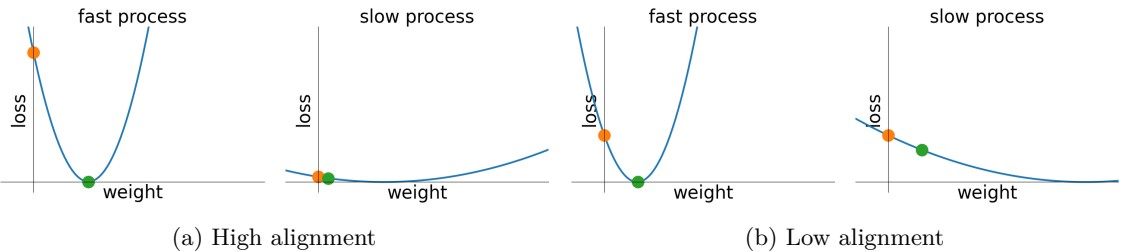

(a) High alignment          (b) Low alignment

Figure 8: The fast and slow processes in a well aligned representation (left) and a poorly aligned representation (right), both with rank 2. The singular values are $\sigma_1 = 6$ for the fast process and $\sigma_2 = 1$ for the slow process. The y-axis is the corresponding share of loss and the x-axis is the projection of weights in the corresponding direction. The orange circle shows the state of the process at zero initialization and the green circle show the state of the process at the end of the initial fast phase, i.e., when the fast process has mostly reduced its loss. With the well aligned representation in (a), the fast process gets a larger portion of overall training loss, and reduces that loss (from orange to green). With low alignment in (b), each process gets half of the overall loss, and so the fast process reduces less of the loss (less change between orange to green). With high alignment, there is little remaining error to reduce in the slow phase. With low alignment, the optimization needs to continue through the slow phase and grow the weights.

We can also consider the optimization process, to understand why convergence is faster. Recall the decomposition of the squared error, in Equation 1, but rewritten slightly, and also now consider its gradient.

$$\|\Phi w - y\|^2 = \sum_{i=1}^{r(\Phi)} \sigma_i^2 \left\| w_i^V - \frac{y_i^U}{\sigma_i} \right\|^2 + \sum_{i=r(\Phi)+1}^{n} \left\| y_i^U \right\|^2,$$

$$\nabla \|\Phi w - y\|^2 = \sum_{i=1}^{r(\Phi)} \sigma_i^2 \nabla \left\| w_i^V - \frac{y_i^U}{\sigma_i} \right\|^2.$$

Each component of the loss will be reduced by movements of the weights in the direction of a right singular vector, seen in the gradient decomposition. Further, the rate of reduction depends on the singular value. We can imagine the weight $w_i^V$ over time as a process, where a larger singular value means larger steps or changes in this process. The processes do not interact with each other since the singular vectors are orthogonal to each other and the movement of the weights in one direction does not depend on its previous movements in other directions. A process with nonzero singular value $\sigma_i$ needs to move the weights to $y_i^U/\sigma_i$ to fully reduce its share of training loss. This required movement is smaller for processes with larger singular values.

From this, we can see that gradient descent will exhibit two phases, as has been previously noted (Oymak et al., 2019a). In an initial fast phase, all the processes are reducing their share of the loss. Since the processes with larger singular values are faster, their loss will be reduced in a few iterations and with a small total movement of the weights. When these processes have essentially stopped, the slower ones with small singular values will keep reducing the loss slowly and growing the weights to large magnitudes. With high alignment gradient descent requires a small magnitude of weights to reduce a large portion of the loss in the initial fast phase. Early stopping after this amount of loss is reduced guarantees small weights, and thus better generalization, without a need for explicitly regularizing the norm of the weights (Oymak et al., 2019a). Figure 8 illustrates the fast and slow processes and the fast and slow phases for two representations with high and low alignment.

## 6.2 A Small Experiment Connecting the Alignment Curve and Convergence Rates

To illustrate the impact of alignment on convergence rates, we fix the representation and create two synthetic label vectors with different alignment properties for that representation. The representation is obtained by sampling 1000 points from the UCI CT Position dataset. The first label vector, $y_1$, is set to the 10th singular

vector (approximately magnitude 70). The second label vector, $y_2$, is a weighted sum of the 5th and the 50th singular vectors (approximately magnitude 115 and 30). We visualize their alignment curves in Figure 9.

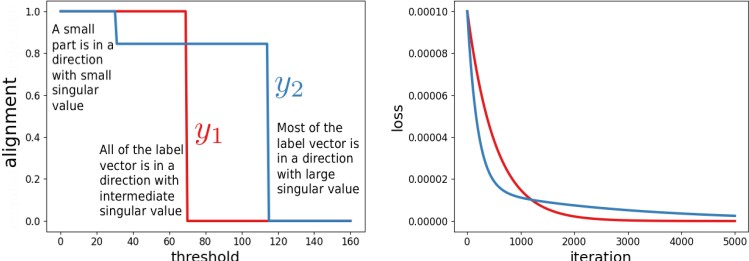

Figure 9: (left) Alignment curves for the two label vectors. After a threshold of 30 the alignment curve drops slightly for $y_2$, representing that a small amount of $y_2$ is aligned with the smaller singular value, and then drops fully at threshold 115, as most of $y_2$ is aligned with the large magnitude singular value. The alignment curve for $y_1$ drops at 70, since all of $y_1$ is aligned with the singular value of magnitude 70. (right) Learning curves for these two label vectors. The convergence rates are as predicted by the alignment curve.

We trained a linear model on each of these two label vectors and plotted the learning curves. These two models in fact show different comparative rates of convergence in early and late training. Initially, the model trained on $y_2$ converges faster as it is reducing the loss corresponding to the 5th singular vector (high magnitude). When this loss is mostly reduced, the model keeps slowly reducing the part that corresponds to the small singular value. The model for $y_1$ consistently decreases error to zero, since it is perfectly aligned with a medium magnitude singular value; it is slower than $y_2$ at first, but then crosses as it maintains the same rate of decrease once $y_1$ enters the slow phase corresponding to reducing error for the small singular value.

## 7 Co-occurrence of Alignment and Transfer

In this section, we discuss why transfer is facilitated when a representation learned on a source task is aligned with the labels for a target task. A representation can be said to facilitate transfer if it improves convergence rates and final accuracy (generalization performance). In the last section, we discussed why alignment can improve convergence rates for a single task. Now, we evaluate more directly in the transfer setting, where the target task differs from the source task.

### 7.1 Positive and Negative Transfer in a Controlled Setting

We demonstrate the correlation between alignment, transfer, and task similarity using peaks functions, a framework of related and unrelated tasks for neural networks (Caruana, 1997). A peaks function is of the form $P_{X,Y,Z} \doteq$ IF $X > 0.5$ THEN $Y$ ELSE $Z$ where $X, Y, Z$ are variables. There are six possible variables A, B, C, D, E, F defined on $[0, 1)$ and the label of a peaks function depends on three out of these six variables. A total of 120 peaks functions can be defined using different permutations of three out of six variables: $P_{A,B,C}, P_{A,B,D}, \dots, P_{F,E,D}$.

Each variable is encoded into 10 features instead of being fed directly to a network as a scalar. The features are obtained by evaluating an RBF function centered at points $0.0, 0.1, \dots, 0.9$, with height 0.5 and standard deviation 0.1. Therefore, there are 60 features representing the 6 variables, and a scalar label that depends on 30 of these features. Inputs are normalized to a length of one and the labels are normalized to have a mean of zero.

The neural network used in the experiments has 60 input units, one hidden layer with 60 neurons and ReLU activation, and one linear output unit. The model is trained by reducing mean squared error using Adam with batch-size 64 and step-size 0.001.

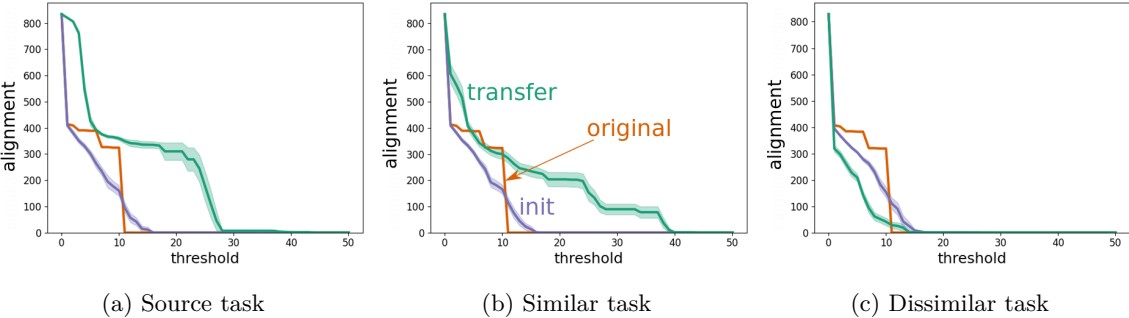

(a) Source task          (b) Similar task          (c) Dissimilar task

Figure 10: Training increases alignment with the source task and related tasks and decreases alignment with unrelated tasks. The plots show alignment between the label vectors of the source task, related task, and unrelated task and the original representation (original), the initial hidden representation (init), and the hidden representation after training on the source task (transfer). The results are averaged over 10 runs. Error bars show standard errors. In (a), training on the source task increases the representation alignment for the source task itself. Alignment with the related task's labels is also increased (b), and the transferred representation is better aligned than the original features. For the unrelated tasks, however, training on the source task reduced the alignment, and the transferred representation is worse than the original features (c).

First we study whether training a neural network on the source task results in a representation that is well aligned with the source tasks and similar tasks. We pick a source task $S$ by randomly choosing 3 variables from A-F and consider two target tasks $T1$ and $T2$. $T1$ has the same variables as $S$ but in a different order and $T2$ uses the three other variables. For example a possible setup is: $S \doteq P_{A,B,C}, T1 \doteq P_{B,A,C}, T2 \doteq P_{E,D,F}$. We create 10000 inputs by sampling the variables A-F randomly from $[0, 1)$ and compute the outputs for the source and target tasks to create a dataset of size 10000 for these tasks. Then we train a neural network only on $S$ until convergence and compare the alignment between the labels of the source and target tasks with the hidden representation after training and the hidden representation at initialization.

The hypothesis is that training on $S$ increases the alignment between the hidden representation and the labels of a task if the task is similar to $S$ and reduces it if the task is dissimilar. We also include the alignment between the original inputs and the labels. The extracted representations are normalized to length one, to appropriately measure alignment. In Figure 10, we can see that the hypothesis is verified: training on the source task increases the alignment for both the source task and the related task, and decreases alignment with the unrelated task.

Next we want to check if representations with better alignment also have better generalization when training data is scarce. This time we choose 100 out of the 10000 points for the related and unrelated task. and compared the performance of a linear model on the original features, hidden representation at initialization, and hidden representation after training on the $S$. The test error is evaluated on a new dataset of size 1000 for evaluating the model's generalization. Figure 11 (a,b) show training and test errors after a certain number of iterations for the related and unrelated task. Figure 11 (c,d) show the norm of weights after a certain reduction of training loss.

The relative performance of the representations mirrors their alignment with the labels. Further, the role of alignment is noticeable in the fast and slow phases of the learning curves. When the target task is similar, the initial fast phase of the transferred representation reduces a large ratio of the overall loss with a small generalization gap and without requiring large weights. Comparison with the representation at initialization shows that this benefit comes from training on the source task. In the unrelated task, where training on the source task reduces alignment, optimization with the transferred features enters the slow phase early in training and performs worse than both the original features and the initial hidden representation. This reduction in alignment provides a clear explanation for this negative transfer.

**Higher layers can specialize to the source task.** We ran the experiment with peaks functions this time with three hidden layers instead of one to let the later layers specialize to the source task. Figure 12 shows

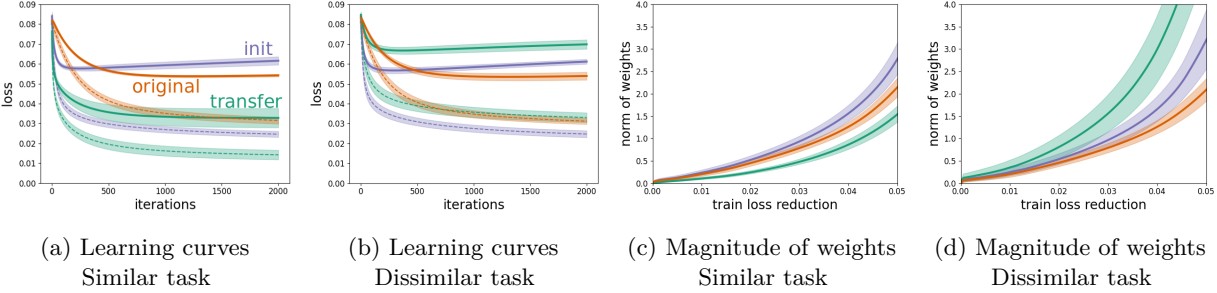

(a) Learning curves
Similar task

(b) Learning curves
Dissimilar task

(c) Magnitude of weights
Similar task

(d) Magnitude of weights
Dissimilar task

Figure 11: Train (dashed lines) and test (solid lines) errors of linear models after a certain number of steps (a,b) and the magnitude of weights after a certain reduction of training loss (c,d). The results are averaged over 10 runs after a sweep on three different step-size values. On the related task (a) for the model with the transferred representation that has better alignment the initial fast phase of the training, where the weights and the generalization gap is small, can substantially reduce the training loss (see the initial sharp drop of the green curve). This fast phase is almost nonexistent in b since the alignment of the transferred representation is decreased, and from the beginning the model needs to grow its weights and the generalization gap to larger magnitudes to reduce the training loss.

layer-by-layer alignment curves of the learned representations on the source task (left) and the related task (right). The monotonous pattern in the main paper persists in the left plot as the representations closer to output adapt to the source task, and there is no such monotonous increase in alignment with another task as seen in the right plot. This observation is consistent with the experiments by Yosinski et al. (2014) on transfer learning with fine-tuning that show layers that are closer to output specialize to the source task, often at the cost of performance on the target task. As the depth of the transferred layer increases, the set of possible target tasks that would be deemed similar to the source task becomes smaller, since the learned representation only keeps the information necessary for the source task.

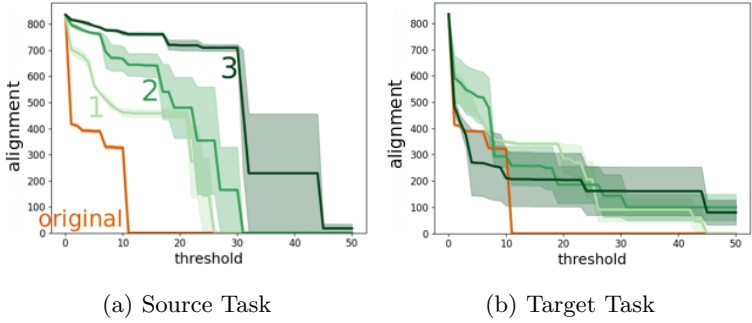

(a) Source Task

(b) Target Task

Figure 12: Layers closer to the output specialize to the source peaks function (left) and are not better for transfer to the target peaks function (right).

## 7.2 Alignment in Pre-trained CNNs and Connection to Transfer

A natural setting to test alignment is in areas where big neural networks are pre-trained for easy re-use. One such setting is Computer Vision, where end-to-end training with deep Convolutional Neural Networks (CNNs) has replaced training linear models on hand-crafted features. Training a deep CNN from scratch, however, requires a large amount of labeled data which is not available for many tasks. Instead, a common strategy is to use transfer (Oquab et al., 2014; Goodfellow et al., 2016). A CNN is trained on a large corpus of data, presumably related to a wide array of target tasks, and made publicly available. The pre-trained model is then used to create features for the target task—often using the last hidden layer of the pre-trained CNN—and a simpler model trained on these features with the more limited dataset for the target task.

We take a CNN first trained on the large ImageNet dataset (Russakovsky et al., 2015), and use features given by the last hidden layer to train a linear model on Cifar10 and Cifar100 (Krizhevsky et al., 2009). The benefit of feature transfer from ImageNet to Cifar10 and Cifar100 is already established in the literature (Kornblith et al., 2019), and we ask whether an increase in representation alignment is behind this gain. The CNN models that are studied are VGG16 (Simonyan & Zisserman, 2014) and the two residual networks ResNet50 and ResNet101 (He et al., 2016).

For each one of target tasks 1000 data points are sampled from two random classes and given $\pm 1$ labels. We then plot the alignment for the input features and the hidden representations before and after training on ImageNet. Recall alignment from equation 2, $\sum_{\{i:\sigma_i \geq \tau\}} (u_i^\top y)^2$ for a given threshold $\tau$. Figure 13 (a) shows the results for ResNet101 on Cifar100. The plots for other architectures and Cifar10 in Appendix C show the same trend.

Training on the source task (ImageNet) significantly increases alignment with the related tasks (Cifar10 and Cifar100). The alignment stays high and drops off suddenly, indicating that the label vector is largely concentrated on larger singular values. An interesting observation is that the hidden representation at random initialization is extremely poorly aligned. The plots for other architectures in the appendix show that the drop in alignment at initialization is more extreme in deeper networks. This mirrors what we observed in Figure 7. A possible explanation is that with more depth, successively more information in the inputs is discarded. Previous work has observed that at initialization, deeper layers have little mutual information with the output (Shwartz-Ziv & Tishby, 2017). Such layers are closer to random vectors, in terms of relationship to the label vector, so unlikely to be well aligned.

We then extract 1000 images from two random classes of Cifar10 and Cifar100 and compare the alignment of representation extracted from ResNet101 pre-trained on ImageNet, SIFT (Lowe, 1999) features with dictionary size 2048, and HOG features (Dalal & Triggs, 2005) with 9 orientations, 64 pixels per cell, and 4 cells per block. We also include Radial Basis Function (RBF) representation of the data with bandwidth 1.0. The input to Radial Basis Functions is a flattened image normalized to length one, and the centers are set to all the 1000 normalized flattened training data points to ensure expressivity. We see in Figure 13 (b) that CNN features show considerably higher alignment than all three other representations on Cifar100. The plot for Cifar10 is in Appendix C.

Finally we train models with both squared error and logistic loss on these representations. Figure 13 (c,d) show the learning curves on Cifar100, and Appendix C provides the plots for Cifar10. With both loss functions, higher alignment results in faster optimization and better performance on test data. An interesting observation is the high training error of RBF despite its expressivity. We computed the closed form least squares solution for this representation and found that it achieves zero squared error and full classification accuracy, which suggests that the high training error of gradient descent is due to extremely slow convergence. Note that alignment is not a simple consequence of linear separability, and, as in the case of this RBF representation, a linearly separable representation can have low alignment.

### 7.3 Co-occurrence of Alignment and Task Similarity in Object Recognition

This section studies the relationship between task similarity and alignment in object recognition tasks with fine-tuning. The first experiment is on transfer between different splits of ImageNet and Cifar10. The second experiment is on the Office-31 dataset and focuses on negative transfer.

**ImageNet-Cifar10:**  We study if pre-training a neural network on a more similar source task in this setting results in higher alignment between the extracted representations and labels of the target task. Yosinski et al. (2014) proposed splitting ImageNet into 551 classes of artificial objects and 449 classes of natural objects for transfer experiments within ImageNet. We adopt this methodology but for transfer experiments from ImageNet to Cifar10. We first split ImageNet into two source tasks of ImageNet-Artificial with 551 classes and ImageNet-Natural with 449 classes. The target tasks are all 6 binary classification tasks between artificial classes of Cifar10 and all 15 binary classification tasks between natural classes of Cifar10. Details of the splits are in Appendix D.

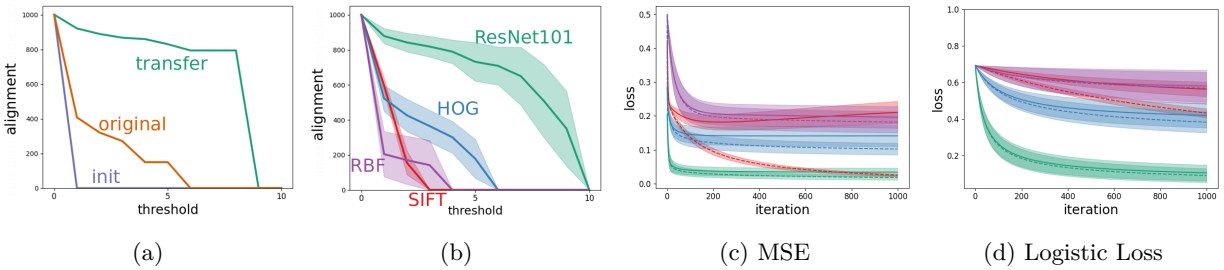

(a)            (b)            (c) MSE          (d) Logistic Loss

Figure 13: Measuring Alignment and Convergence rate on Cifar100 using a ResNet101 Pre-Trained on ImageNet. (a) Representation alignment for input features (original), hidden representation at initialization (init), and hidden representation after training on ImageNet (transfer). Training on ImageNet increases the alignment. (b) ResNet features show higher alignment than handcrafted image features and RBF features. The curves are averaged over 5 runs and the shades show standard errors. (c,d) Train (dashed) and test (solid) learning curves for squared error and logistic loss minimization on the representations in the second plot. CNN representations show the best performance.

The hypothesis is that pre-training on ImageNet-Artificial results in higher alignment than pre-training on ImageNet-Natural if the target task is classification between artificial objects and lower alignment if the target task is classification between natural objects. The architectures in this experiment are ResNet18 and Tokens-to-Token Vision Transformer (T2T-ViT) (Yuan et al., 2021). We first pre-train the model on each source task. Then we create a balanced dataset of size 2000 from two natural or two artificial classes of Cifar10, feed the input of this dataset to the two pre-trained networks, and extract hidden representations from the last four hidden layers. Each of these extracted representations results in an alignment curve. For each layer, if the target task is classification between natural objects, we subtract the alignment curve of the model pre-trained on ImageNet-Artificial from the alignment curve of the model pre-trained on ImageNet-Natural. If pre-training on ImageNet-Natural results in higher alignment for this target task, the result of this subtraction will be positive. For a target task of classification between artificial objects, we subtract the alignment curve of the model pre-trained on ImageNet-Natural from the alignment curve of the model pre-trained on ImageNet-Artificial. Again, the hypothesis is corroborated if the result of this subtraction is positive.

Figure 14 shows the results of the subtraction averaged over the 21 target tasks to minimize clutter. Results for individual target tasks are provided in Appendix D. On average over the target tasks, pre-training on a similar source task results in higher alignment in all the studied hidden layers and across all values of threshold in ResNet18 and a wide range of thresholds in T2T-ViT. For ResNet18, the difference was more extreme for layers closer to the output, indicating that the hidden layers closer to the output have specialized to the source task.

We also ask if this increase in alignment correlates with transfer performance in a more common setup of fine-tuning a multi-class classification network. Similar to ImageNet, we split Cifar10 to two target tasks of Cifar10-Natural with six classes and Cifar10-Artificial with four classes and fine-tune each pre-trained network on each target task. As seen in Table 15, for both target tasks and with both architectures pre-training on the similar source task, which has better alignment in the corresponding binary classification tasks, also results in better fine-tuning performance in multi-class classification. Studying the relationship between alignment in hidden layers and dynamics of a fine-tuned multi-class network is beyond the scope of this paper and is left for future work.

**Office-31:** We test whether negative transfer correlates with a reduction in alignment in an experiment inspired by the recent study on negative transfer (Wang et al., 2019). The dataset is Office-31 (Saenko et al., 2010) which has three domains Amazon, Webcam, and DSLR. The images in these domains are taken with different types of cameras and this results in a mismatch between the input distributions. The task is to classify 31 classes and the classes are shared among the domains. The neural network is a ResNet18 which has been pre-trained on ImageNet before the experiment, as otherwise all final accuracies would be low. Each time, one domain is picked as the source task and its labels are replaced with uniform random numbers

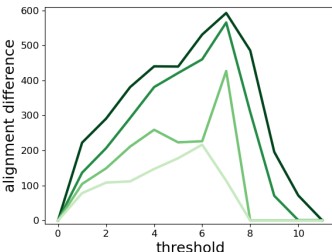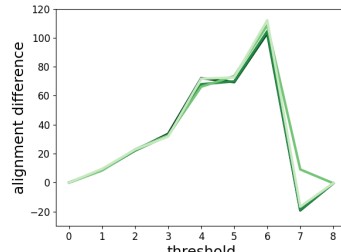

Figure 14: Results of subtracting alignment curves for the last four hidden layers of ResNet18 (left) and T2T-ViT (right), averaged over the target tasks. Each curve corresponds to a layer and darker means closer to output. For each layer and each individual binary classification task, we subtracted the alignment curve of the model pre-trained on the less similar source task from the one obtained from pre-training on the more similar source task. Note the y-axis is different in the plots. All the four subtracted curves have large positive numbers across all thresholds for ResNet18 and a wide range of thresholds for T2T-ViT.

| Source \Target | Natural | Artificial | Source \Target | Natural | Artificial |
|---|---|---|---|---|---|
| Natural | **94.55** | 95.70 | Natural | **84.33** | 91.00 |
| Artificial | 90.49 | **97.14** | Artificial | 83.10 | **92.80** |

Figure 15: Multi-class classification test accuracy of ResNet18 (left) and T2T-ViT (right) in percent after fine-tuning. Each row is a source task from ImageNet and each column is a target task from Cifar10. For both target tasks, pre-training on similar source task resulted in better performance. This mirrors the relative alignments in corresponding binary classification tasks.

and a different domain is picked as the target task. The difference in input distribution together with label randomization in the source task results in negative transfer to the target task, that is, a network that is pre-trained on the source task and then fine-tuned on the target task performs worse than a network that is directly trained on the target task. Table 1 verifies negative transfer in all pairs of source and target task in multi-class classification. Pre-training on Amazon and DSLR resulted in the largest and smallest degree of negative transfer. Further experiment details are in Appendix D.

For each pair of source and target domain and for each binary classification task in the target domain we subtract the alignment curve of the model that is pre-trained on the souce task from the alignment curve of the model that is not pre-trained on the source task. The alignment curves are computed from the last hidden layer before fine-tuning and, similar to the previous experiment, the differences are averaged over the binary classification tasks. If the difference is positive it means that, on average over the binary classification tasks, pre-training on the source task has reduced alignment for the target task. The results are provided in Figure 16 where each curve corresponds to a pair of source and target task. The curves all have positive values which shows a reduction in alignment in all the six pairs of tasks. Furthermore, pre-training on Amazon and DSLR results in largest and smallest reduction in alignment, which mirrors the degree of negative transfer in Table 1.

## 8   Related Work

Early work by Hansen (1990) studied the closed-form solution of linear regression with $\ell_2$ regularization or truncated SVD under the Discrete Picard Condition (DPC). DPC states that the magnitude of projections of the label vector on the left singular vectors decays to zero faster than the sequence of singular values. A follow-up paper (Hansen, 1992) studied the closed-form solution of $\ell_2$-regularized regression under DPC using L-curves which show the magnitudes of residual and weights at different levels of regularization. The name of these plots comes from their L-shaped appearance which means that a certain amount of residual can be reduced with small weights and reducing the rest of the loss requires large weights. The authors attributed this pattern to the projection of the label vector onto the directions of large and small singular values.

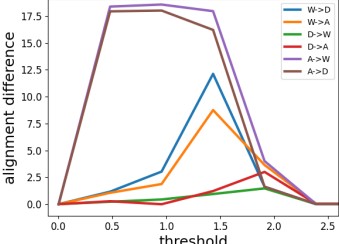

Figure 16: Results of subtracting alignment curves for the last hidden layer of ResNet18 in the Office-31 dataset, averaged over the binary classification tasks in the target domain. Each curve corresponds to a pair of source and target domain. The differences are largely positive which means that pre-training on the source task reduced alignment in all these cases.

| Source \Target | DSLR | Webcam | Amazon |
|---|---|---|---|
| None | **88.0** | **92.5** | **82.7** |
| DSLR | - | 70.3 | 65.6 |
| Webcam | 30.7 | - | 47.3 |
| Amazon | 4.7 | 4.6 | - |

Table 1: Test accuracy in percent after fine-tuning in the Office-31 dataset. Each row is a source task and each column is a target task. The first row, None, shows the performance of the model that was not pre-trained on any of these source tasks. In all pairs of source and target tasks, pre-training considerably reduced performance. This mirrors the relative alignments in corresponding binary classification tasks.

Understanding neural networks surprising generalization ability is an active area of research (Jiang et al., 2019). Neural networks generalize well on real-world data despite having the capacity to fit random labels Zhang et al. (2017). This phenomenon, called benign overfitting, has been studied in linear regression and associated with the eigenspectrum of the covariance matrix (Bartlett et al., 2020). Work on the double-descent phenomenon (Belkin et al., 2019; Nakkiran et al., 2021) has also challenged the classic notion of bias-variance trade-off and provided a framework for studying generalization of models with high capacity.

Jacot et al. (2018) provided the Neural Tangent Kernel (NTK) framework that approximates the behaviour of overparametrized neural networks and can be studied theoretically. They observed faster convergence of neural networks in directions that correspond to the first eigenvectors of the kernel's Gram matrix. As label randomization tests by Zhang et al. (2017) resurrected interest in the relationship between the task and representation, Arora et al. (2019) studied the difference between datasets with true and random labels for gradient descent optimization and generalization. They provided an analysis for overparameterized neural networks that requires projecting the label vector on the eigendirections of the Gram matrix and found that true labels are mostly aligned with directions of large singular values. Oymak et al. (2019a) refined the analysis of Arora et al. (2019) by separating the large and small eigenvalues with a threshold and provided better generalization guarantees by early stopping before the slow processes grow the weights to large magnitudes.

Recent work has explored adaptation of the network to a task (Oymak et al., 2019a;b; Baratin et al., 2020; Kopitkov & Indelman, 2020; Ortiz-Jiménez et al., 2021). They showed that the matrix of a kernel associated with the network evolves such that the labels are mostly aligned with the first eigenvectors of the corresponding matrix. Through a similar functional view, Lampinen & Ganguli (2019) studied adaptation of deep linear networks to the task without the kernel approximation and showed the benefit of learning multiple related tasks at the same time.

It is important to distinguish this form of adaptation, which is sometimes called "neural tangent feature alignment" or "neural feature alignment," and alignment of hidden representations. Neural tangent feature alignment views the whole network as a black-box function and studies adaptation of the model in the space of functions. Our work investigates the intermediate layers one by one and shows that each one increases alignment in its representation. For transfer learning, neural tangent feature alignment implies that if a network is trained on a source task, the whole network can be further trained efficiently on a related target task. Hidden representation alignment means that the features learned in each hidden layer are suitable for training a new model on the related target task.

Finally, Maennel et al. (2020) showed how the weights in each layer of neural network adapt their spectral properties to the input features, which explains the benefit of feature transfer on tasks with similar input. This analysis disregards adaptation to labels and cannot explain why transfer to unrelated tasks on the same input can hurt (like the case of negative transfer to dissimilar peaks functions). The authors explain negative transfer through inactivity of ReLU units.

## 9 Conclusion

In this work, we developed the concept of representation alignment in neural networks and showed that it correlates with transfer performance. This alignment emerges from the training process. In contrast, we showed that random neural networks have poor alignment, even if they are sufficiently large to enable accurate approximation. We measured the properties of a variety of neural network representations, showing that it correlates with positive transfer for similar tasks and negative transfer for dissimilar tasks; that widely used CNNs have high levels of alignment; and finally that this property appears to emerge across a variety of architectures, with higher alignment in the later layers.

Understanding how the dynamics of gradient descent leads to the emergence of alignment is an important next step. Recent work on the Neural Collapse (NC) phenomena (Papyan et al., 2020) can help with this direction. The first of these phenomena (NC1) is that at late stages of training, within-class variations of the last hidden layer activations shrink to zero and the activations collapse to the class means. Going back to the example in Figure 2 we can see the relationship to alignment. In the left figure where the points in each class are close to each other alignment is high. In the figure on the right, where the points belonging to the same class are separated along the top principal component and are therefore further apart, the label vector is mostly in the direction with smaller singular value and alignment is low. Although this argument is still preliminary and the work on NC focuses on the last hidden layer and late training, we still highlight this relationship to provide a starting point for future theoretical work.

**Acknowledgment:** The authors would like to thank Alireza Fallah, Behnam Neyshabur, members of the RLAI Lab at the University of Alberta, and the anonymous reviewers for their invaluable comments and discussions through this project. This research is funded by the Canada CIFAR AI Chairs program and Alberta Machine Intelligence Institute.

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

## A    Other Plots for Measuring Alignment in Different Neural Networks

This plots in Figures 17 and 18 show the increase in alignment on training data with different optimizers and batch-sizes as well as the increase in alignment on test data.

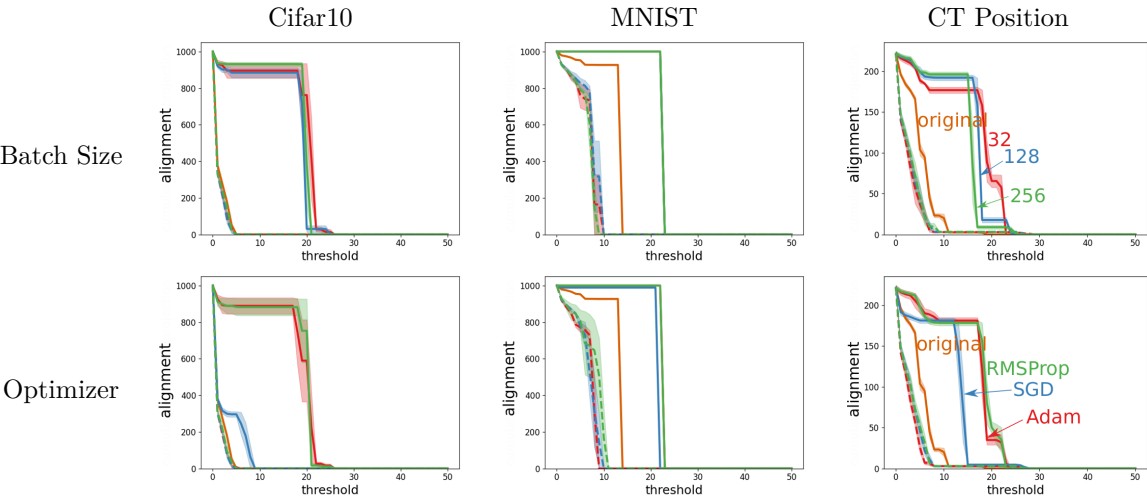

Figure 17: Training increases alignment in hidden representations regardless of batch-size and optimizer. The orange line is the input features, the dashed lines are hidden representations before training, and the solid lines are hidden representations after training. The plots are averaged over 5 runs and the error bars show standard errors. The label vector for the regression task was moved to the interval $[-1, 1]$ for easier comparison with classification. The curves for learned representation on MNIST mostly overlap as all models can achieve a high level of alignment on this task.

## B    Convergence Rate

For completeness, we include the convergence rate results here.

**Proposition 1.** *Given full rank representation matrix $\Phi$ and label vector $y$, let $w^* = (\Phi^\top \Phi)^{-1} \Phi^\top y$ be the optimal linear regression solution and $\sigma_{max}$ be the maximum singular value for $\Phi$. The batch gradient descent updates $w_{t+1} = w_t - \eta \Phi^\top(\Phi w_t - y)$ with stepsize $0 < \eta < \sigma_{max}^{-2}$ and $w_0 = 0$ results in*

$$\|\Phi w_t - \Phi w^*\| = \sqrt{\sum_{j=1}^{r(\Phi)} (1 - \eta \sigma_i^2)^{2t} (u_i^\top y)^2} \tag{4}$$

$$\|\Phi w_t - y\|^2 = \sum_{j=1}^{r(\Phi)} (1 - \eta \sigma_i^2)^{2t} (u_i^\top y)^2 + \|\Phi w^* - y\|^2$$

*Proof for Proposition 1.* For $\Phi = U \Sigma V^\top$ the thin SVD of $\Phi$ with $\Sigma \in \mathbb{R}^{n \times d}$, let $\Lambda = \Sigma^\top \Sigma$, $A = \Phi^\top \Phi = V \Lambda V^\top$ and $b = \Phi^\top y = V \Sigma^\top U^\top y$. The gradient descent update corresponds to the following iterative linear system update

$$w_{t+1} = w_t - \eta \Phi^\top(\Phi w_t - y) = w_t - \eta(A w_t - b)$$
$$= (I - \eta A) w_t + \eta b$$

Starting from $w_0$, we get that

$$w_1 = \eta b = \eta (I - \eta A)^0 b$$
$$w_2 = (I - \eta A)(\eta b) + \eta b = \eta[(I - \eta A)^0 + (I - \eta A)] b$$
$$\cdots$$
$$w_{t+1} = \eta \sum_{i=0}^{t} (I - \eta A)^i b$$

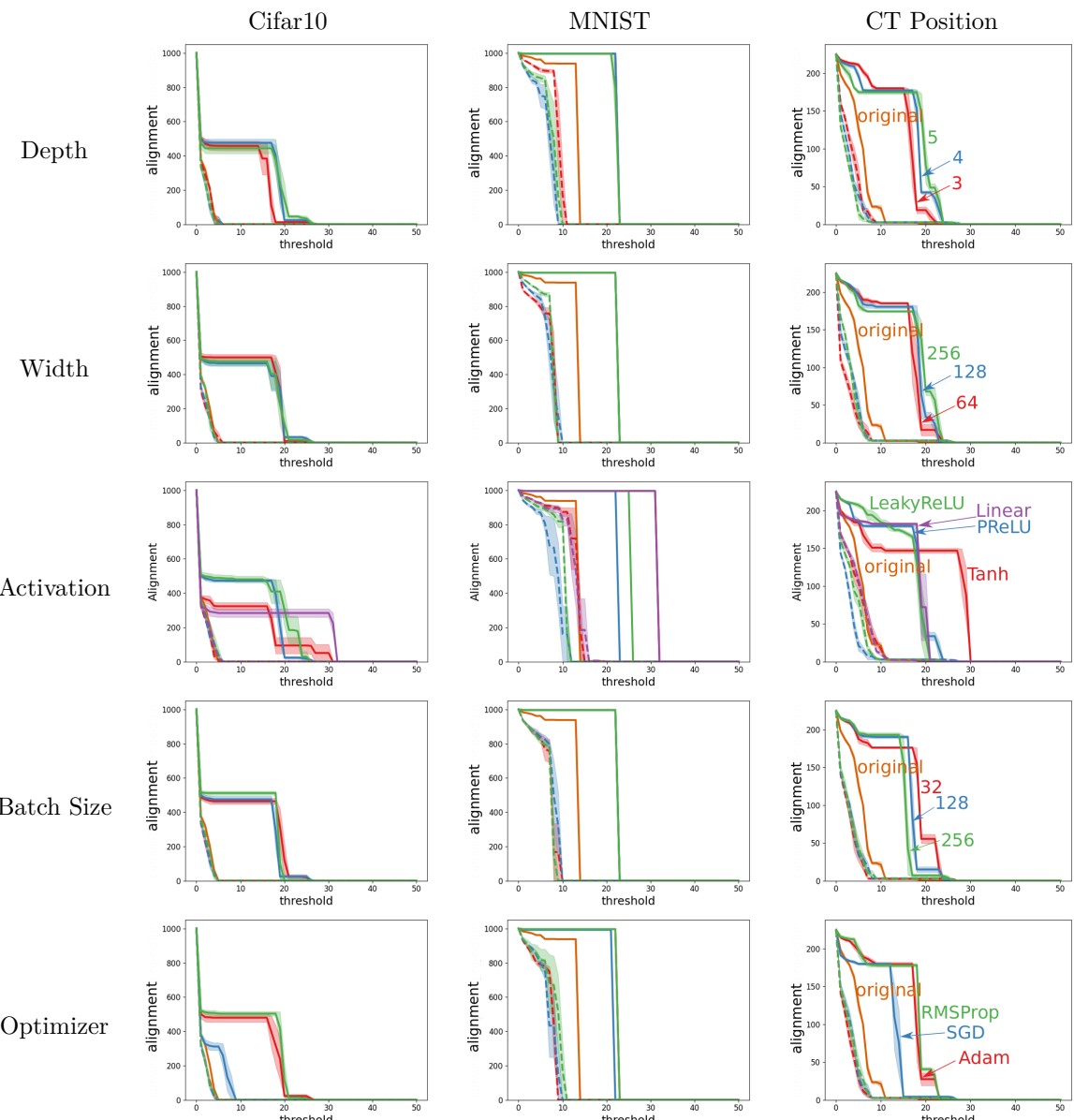

Figure 18: Training increases alignment in hidden representation on test data. The orange line is the input features, the dashed lines are hidden representations before training, and the solid lines are hidden representations after training. The plots are averaged over 5 runs and the error bars show standard errors. The label vector for the regression task was moved to the interval $[-1, 1]$ for easier comparison with classification.

Further,

$$I - \eta A = V^\top V - \eta V \Lambda V^\top = V(I - \eta \Lambda)V^\top$$

$$(I - \eta A)^i = V(I - \eta \Lambda)^i V^\top$$

$$\sum_{i=0}^{t}(I - \eta A)^i = V \sum_{i=0}^{t}(I - \eta \Lambda)^i V^\top$$

If we let $\tilde{\Lambda}_t \doteq \sum_{i=0}^t (I - \eta\Lambda)^i$ and $\lambda_j$ the $j$ entry on the diagonal in $\Lambda$, then because $\lambda_j > 0$ for all $j$, each component of this diagonal matrix corresponds to

$$\tilde{\Lambda}_{t,j} = \sum_{i=0}^t (1 - \eta\lambda_j)^i = \frac{1 - (1 - \eta\lambda_j)^{t+1}}{1 - (1 - \eta\lambda_j)} = \frac{1 - (1 - \eta\lambda_j)^{t+1}}{\eta\lambda_j}$$

Additionally,

$$(I - \eta A)^i b = V(I - \eta\Lambda)^i V^\top V \Sigma^\top U^\top y$$
$$= V(I - \eta\Lambda)^i \Sigma^\top U^\top y$$

For $\tilde{\Lambda}_{*,j} = (1 - \eta\lambda_j)^{-1} = \frac{1}{\eta\lambda_j}$, where the inverse exists due to conditions on $\eta$, we have $w^* = \eta V \tilde{\Lambda}_* \Sigma^\top U^\top y$. Finally we get

$$\|w_{t+1} - w^*\| = \eta \left\| V\Lambda_t \Sigma^\top U^\top y - V\tilde{\Lambda}_* \Sigma^\top U^\top y \right\|$$
$$= \eta \left\| V(\tilde{\Lambda}_t - \Lambda_*)\Sigma^\top U^\top y \right\|$$
$$= \eta \left\| (\tilde{\Lambda}_t - \Lambda_*)\Sigma^\top U^\top y \right\|$$

Notice that $\tilde{\Lambda}_{*,j} - \tilde{\Lambda}_{t,j} = \frac{(1-\eta\lambda_j)^{t+1}}{\eta\lambda_j}$ and each component of $\eta(\Lambda_* - \tilde{\Lambda}_t)\Sigma^\top$ equals $\frac{(1-\eta\lambda_j)^{t+1}}{\sigma_j}$. We therefore get

$$\|w_{t+1} - w^*\|^2 = \left\| \eta(\Lambda_* - \tilde{\Lambda})\Sigma^\top U^\top y \right\|^2$$
$$= \sum_{j=1}^{r(\Phi)} \left[ \frac{(1 - \eta\lambda_j)^{t+1}}{\sigma_j} u_j^\top y \right]^2$$
$$= \sum_{j=1}^{r(\Phi)} (1 - \eta\sigma_j^2)^{2(t+1)} \frac{(u_j^\top y)^2}{\sigma_j^2}$$

which shows the first part of the proposition. The alignment relationship is clear in this result. The term $\frac{(u_j^\top y)^2}{\sigma_j}$ is big if $(u_j^\top y)^2$ is big and $\sigma_j$ is small. This is further exacerbated by the fact that $(1 - \eta\sigma_j^2)^{2(t+1)}$ shrinks slowly if $\sigma_j$ is small. On the other hand, consider the case that $(u_j^\top y)^2$ is notably smaller than $\sigma_j$, even if $\sigma_j$ is small. Then this term contributes only a small amount to the loss in the beginning, i.e., $\frac{(u_j^\top y)^2}{\sigma_j}$, and this small amount is then reduced with each iteration, slowly if $\sigma_j$ is small and quickly if $\sigma_j$ is big. The ideal rate is obtained when high magnitude $(u_j^\top y)^2$ is concentrated on large $\sigma_j$. Now we show the rate of convergence of the predictions:

$$\|\Phi w_{t+1} - \Phi w^*\|^2 = \|\Phi(w_{t+1} - w^*)\|^2$$
$$= \left\| \eta U\Sigma V^\top V(\Lambda_* - \tilde{\Lambda})\Sigma^\top U^\top y \right\|^2$$
$$= \left\| \eta U\Sigma(\Lambda_* - \tilde{\Lambda})\Sigma^\top U^\top y \right\|^2$$
$$= \left\| \eta\Sigma(\Lambda_* - \tilde{\Lambda})\Sigma^\top U^\top y \right\|^2$$
$$= \sum_{j=1}^{r(\Phi)} \left[ (1 - \eta\lambda_j)^{t+1} u_j^\top y \right]^2$$
$$= \sum_{j=1}^{r(\Phi)} (1 - \eta\sigma_j^2)^{2(t+1)} (u_j^\top y)^2$$

The term $(1 - \eta\sigma_j^2)^{2(t+1)}$ appears again in this result and shows the improved convergence rate when $(u_j^\top y)^2$ is concentrated on directions of large singular values.

Now we can relate convergence in predictions to convergence in loss. Gradient descent under the conditions of the previous proposition converges to the minimum squared error solution. Define $l_w \doteq \|\Phi w - y\|^2$ for

any $w \in \mathbb{R}^d$. If $l_{w^*} = 0$ then clearly $l_w = \|\Phi w - y\|^2 = \|\Phi w - \Phi w^*\|^2$. Otherwise $\Phi w^*$ is the projection of $y$ on the span of $\Phi$, and $\Phi w^* - y$ is perpendicular to any vector within this subspace, including $\Phi w - \Phi w^*$. Pythagorean theorem then gives $\|\Phi w - y\|^2 = \|\Phi w^* - y\|^2 + \|\Phi w - \Phi w^*\|^2$. $\qquad\square$

**Proposition 2.** *Under the conditions of the previous proposition, if $Alignment(\Phi, y, \tau) = \delta$ for a threshold $0 < \tau \leq \sigma_{max}$, then gradient descent needs at most $-\log(1 - \omega/\delta)/(2\eta\tau^2)$ iterations to reduce the loss by $0 \leq \omega < \delta$.*

*Proof for Proposition 2.* The amount of reduction in loss after $t$ iterations is denoted by $\omega$ and is equal to

$$\omega \doteq l_{w_0} - l_{w_t} = \|\Phi w_0 - \Phi w^*\|^2 - \|\Phi w_t - \Phi w^*\|^2$$

$$= \sum_{j=1}^{r(\Phi)} \left(u_j^\top y\right)^2 - \sum_{j=1}^{r(\Phi)} (1 - \eta\sigma_j^2)^{2t} \left(u_j^\top y\right)^2$$

$$= \sum_{\sigma_j \geq \tau} \left(u_j^\top y\right)^2 - (1 - \eta\sigma_j^2)^{2t} \left(u_j^\top y\right)^2 + \sum_{0 < \sigma_j < \tau} \left(u_j^\top y\right)^2 - (1 - \eta\sigma_j^2)^{2t} \left(u_j^\top y\right)^2$$

$$\geq \sum_{\sigma_j \geq \tau} \left(u_j^\top y\right)^2 - (1 - \eta\sigma_j^2)^{2t} \left(u_j^\top y\right)^2 + 0$$

$$\geq \delta - (1 - \eta\tau^2)^{2t}\delta$$

where the second last step follows because each term in the second sum is non-negative and the last step follows from the definition of alignment. Now, due to the condition on the step-size and for $0 < \tau \leq \sigma_{max}$ we have $0 < \eta\tau^2 < 1$ and therefore $(1 - \eta\tau^2)^{2t} < \exp(-2t\eta\tau^2)$. Therefore

$$\omega \geq \delta(1 - \exp(-2t\eta\tau^2)) \implies \exp(-2t\eta\tau^2) \geq 1 - \omega/\delta \implies -2t\eta\tau^2 \geq \log(1 - \omega/\delta)$$
$$\implies t \leq -\log(1 - \omega/\delta)/(2\eta\tau^2)$$

$\qquad\square$

## C   Other plots for Pre-Trained CNNs

Figure 19 shows alignment plots for different pre-trained CNN architectures. Figure 20 compares alignment curves of CNN features and handcrafted features on Cifar10.

## D   Fine-Tuning Experiment Details

**ImageNet-Cifar10 Splits:**   The classes in artificial and natural splits of ImageNet are the ones reported by Yosinski et al. (2014). Cifar10-Natural consists of classes bird, cat, deer, dog, frog, horse, and Cifar10-Artificial includes airplane, automobile, ship, and truck.

**ImageNet-Cifar10 ResNet18 Hyperparameters:**   Optimizer: SGD; Momentum: 0.9; Batch-Size: 1024 (distributed over 4 V100 GPUs) for pre-training and 256 (on 1 K80 GPU) for fine-tuning; Initial Learning Rate: 0.1 for pre-training and 0.01 for fine-tuning; Learning Rate Decay Schedule: multiplied by 0.1 after each 30 epochs; Total Number of Epochs: 40 for pre-training and 10 for fine-tuning; Weight Decay: 1e-4; Image Transformations: resize to 256x256 and random cropping and horizontal flipping augmentation.

**ImageNet-Cifar10 T2T-ViT Hyperparameters:**   Optimizer: SGD; Momentum: 0.9; Batch-Size: 256 (distributed over 2 V100 GPUs) for pre-training and 64 (on 1 V100 GPU) for fine-tuning; Initial Learning Rate: 0.01; Total Number of Epochs: 30 for pre-training and 10 for fine-tuning; Weight Decay: 1e-4; Image Transformations: resize to 256x256 and random cropping and horizontal flipping augmentation.

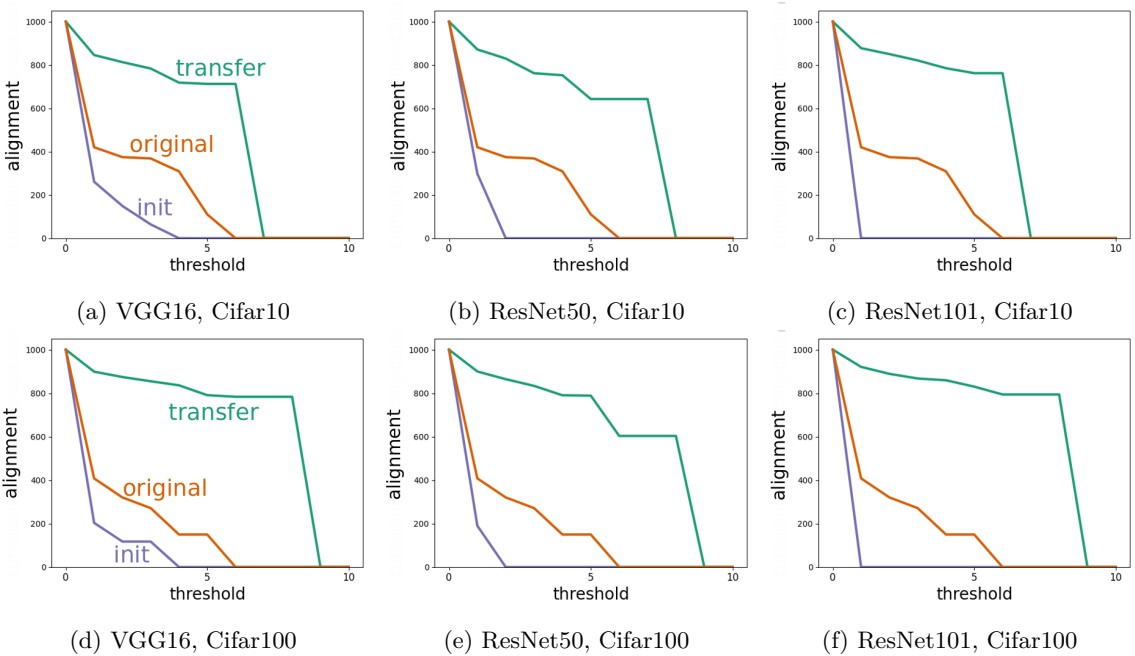

Figure 19: Representation alignment for input features (original), hidden representation at initialization (init), and hidden representation after training on ImageNet (transfer). Training on ImageNet increases the alignment for all three models and on both Cifar10 and Cifar100.

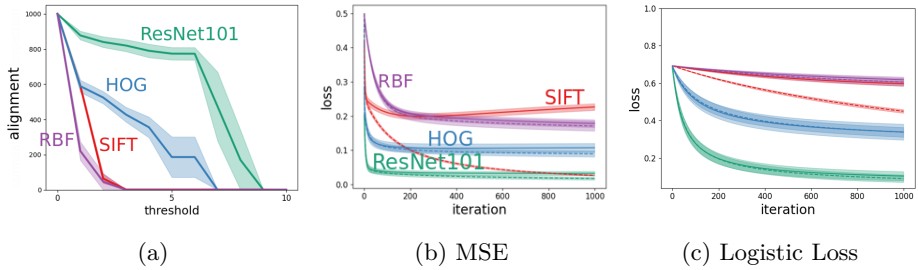

Figure 20: Measuring Alignment and Convergence rate on Cifar10 using a ResNet101 Pre-Trained on ImageNet.(a) ResNet features show higher alignment than handcrafted image features and RBF features. The curves are averaged over 5 runs and the shades show standard errors. (b,c) Train (dashed) and test (solid) learning curves for squared error and logistic loss minimization on different representations. CNN representations show the best performance.

**Office-31 Hyperparameters:** Optimizer: Adam for pre-training and SGD with momentum 0.9 for fine-tuning; Batch-Size: 256 (on 1 V100 GPU) for pre-training and 256 (on 1 V100 GPU) for fine-tuning; Initial Learning Rate: 0.001; Learning Rate Decay Schedule: multiplied by 0.1 after each 30 epochs; Total Number of Epochs: 500 for pre-training and 100 for fine-tuning; Weight Decay: 1e-4; Image Transformations: resize to 256x256 and random cropping and horizontal flipping augmentation.

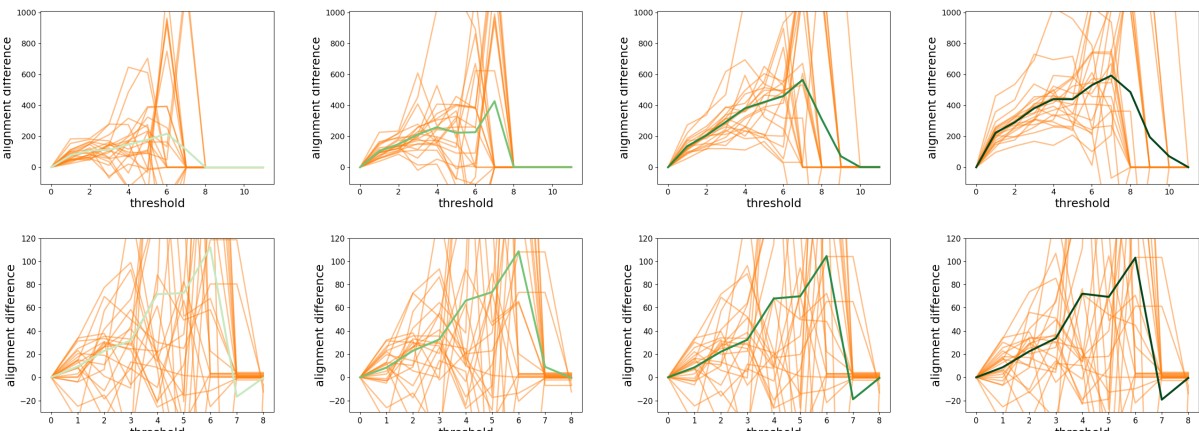

Figure 21: Individual alignment difference curves for the 21 binary classification target tasks with ResNet18 (above) and T2T-ViT (below). Each plot corresponds to one hidden layer. The orange and green curves show individual curves and the averaged curves reported in the main paper.

