# OpenReview forum: "Representation Alignment in Neural Networks"
_TMLR — Accepted by TMLR_

### Review · Reviewer_7c5a · 2022-06-24

**Summary Of Contributions:**

- At the highest level, the paper conducts an empirical study based on a notion of representation alignment that is associated with the alignment of a source/target task (in terms of labels) and directions of highest variation in the representation space (as identified by singular vectors of observed data).
- Using this notion of alignment, it makes a number of empirical observations regarding the learnt representations of neural networks at the final layer of trained deep networks as follows.
- The representations become more aligned after training a deep network for the source task.
- The deeper representations of a trained network are increasingly more aligned for the source task.
- The representation learnt by a deep network is more aligned than those learnt by RBF networks or unsupervised sparse coding optimizations.
- In the context of transfer learning, poorly-aligned representations can explain negative transfers for a synthetic regression task.
- In the context of transfer learning, the alignment correlates with similarity of image classification tasks as measured by semantic tree distances (wordnet).


**Broader Impact Concerns:**

The reviewer does not see any direct necessity for the inclusion of such section in this specific paper.

**Requested Changes:**

The main weaknesses I see that need to be addressed are regarding the conclusivity of the experiments, the clarity, and the discussion and comparison with other related works as outlined above.

**Strengths And Weaknesses:**

**Strengths**
+ The empirical study is fairly thorough and covers many aspects of learning deep representations and spans different setups for the architecture, optimization, and tasks.
+ The results are generally consistent and mostly conclusive for the different setups and tasks that are considered.
+ Predicting and explaining negative transfer is an open and important direction which seems to be correlated with the used empirical notion.

**Weaknesses**

*Results and Experiments*
- the paper sometimes implies causal relation between the representation alignment and two notions of generalization and convergence. It should be discussed clearly if it is causality or correlation that the paper is after for each of those two properties and then accordingly discuss if the experiments and observations are sufficient for either of the cases.
- for generalization, the correlation between generalization and alignment is shown through improved accuracy. One important question here is the case of overfitting. Does the alignment increase during overfitting? If so, does this affect the hypothesis on generalization? How?
- for convergence, the provided empirical evidence for improved convergence is quite limited. A conclusive observation requires more (well-defined) experiments in different setups and ideally accompanied by experiments on “real-world” datasets that clearly isolate the convergence rate from other factors. Some controlled setup such as Figure 9 are pedagogical and interesting and are important to include as well but even there it needs more in-depth study to establish that the discussed trend in convergence is in fact the result of the design (with the aligned non-zero singular values).
- the results regarding negative transfer are quite interesting but the task is limited for a general claim w.r.t. alignment. Can this be extended to some other controlled but more realistic learning tasks?

*Related works*
- the prior empirical and theoretical findings regarding (some defined notions of) representation margin between classes in the intermediate representation space of trained neural networks should have some (potentially direct) relevance to the notion of alignment as used in this work, especially when it comes to correlating it with generalization. This is largely missing from the paper in different sections. The related literature is rich but one possible starting point to find such relevant papers could be the following work, the ones it cites and the ones that cites it:
“Fantastic Generalization Measures and Where to Find Them” 2019.

*Clarity*
- the paper commonly states that the deep network’s representation for this study is taken from the last layer. Is the last layer the prediction layer or the last “hidden” layer which is the one before the outputs? If it is the last prediction layer, for the case of classification, is it the logits or after applying the logistic function? How does this affect the discussions around the observations?
- on the one hand the whole empirical study is based on the comparison of “alignment” in different setups and contrasting high and low alignment, on the other hand, a precise definition of high and low alignment is generally missing. The definition of alignment in equation 2 only considers alignment for a single threshold. The discussions throughout the paper, however, use a qualitative notion of alignment that is never precisely defined based on comparing the plots of the defined alignment over different thresholds. Although the figures suggest a clear qualitative separation in most cases, this can become a source of confusion and subjective analysis and needs to be rectified. That is, hypothetically, not all cases will have one plot completely above or below another; they can intersect in general.

*Minor points*
- page 2: “Saliency maps and Layerwise Relevance Propagation”: citations missing
- page 2: “n>=d”: is this always true? Especially for the target task. A discussion is missing on the necessity and imposed limitations of this condition.
- page 3: when referring to Figure 2a, it does not become clear until later where the representation $\Phi$ is taken from.
- page 1-4: generally “alignment” is taken as an accepted notion and phrases such as “alignment is” are generally used. It might be better to use phrases such as “in this work alignment is defined as” or provide references to prior works to establish that this is *the* accepted definition of representation alignment in the literature.
- page 4: “figure 19” -> probably meant to refer to the main paper’s figure 4
- page 5: “Equation equation”
- page 6: “increases alignment” -> increase alignment
- page 6: “help speed learning”
- fig 6, caption: “error bars” -> probably “shaded area”
- fig 6, caption: standard error or standard deviation?
- fig 6: the orange/red plot for the “original” and the orange/red for the trained alternatives are hard to distinguish and causes confusion since “original” is an important reference.
- page 7: “the aim of this paper paper is show” -> to show / showing

---

> ### Author Response · Authors · 2022-07-08
> **Response**
>
> We really appreciate the time you put on this thorough review and these helpful comments.
>
> Correlation or causal relationship: Thank you for pointing this out. Our goal is to present results on co-occurrence of alignment and transfer along with motivating discussion and arguments. To ensure consistency between claims and evidence, we edited statements that suggested a causal relationship in the revision.
>
> Overfitting: We suspect that in case of overfitting alignment increases for the training data but not for test data. For example, in Figure 2 we train on random labels. The networks memorizes the random labels and alignment in the last hidden representation increases for this particular training set. In our experiments this increase in alignment did not generalize to a separate test set with random labels. (We can add the plot to the paper if you think it is necessary.) This is expected as the network has only memorized the training labels and there is no shared pattern between the training set and test set's labels. For experiments in Figure 6, we have alignment curves evaluated on a test set in the appendix. With these datasets and architectures, the network increases alignment on both training and test data.
>
> Negative Transfer Experiments: Understanding negative transfer was one of our driving forces in this research. The revision includes an experiment on six tasks of the Office-31 dataset with different degrees of negative transfer which correlate with reduction in alignment.
>
> Related Work: We have cited the mentioned paper along with recent work on double descent and benign overfitting in the revision. The literature on generalization is vast and we can only cover so much. Please let us know if there are other papers that are particularly relevant to this research.
>
> Last layer: This is the last hidden layer's activations. We clarified this in the revision.
>
> Intersecting alignment curves: Alignment curves can intersect in general, and this has a specific effect on convergence rate of gradient descent on these representations. The aim of Section 6.2 in our paper is to construct this situation and demonstrate its effect. The blue curve in this experiment that has higher alignment (compared to the red curve) with large thresholds and lower alignment with small thresholds has faster convergence rate (compared to the red curve) in the beginning of training and lower convergence rate at the end of training. To account for these subtleties and provide a more thorough picture, we insist on reporting an alignment curve (across thresholds) in our experiments instead of reducing it to a scalar measure. We are careful to not claim an increase in alignment unless it holds across a large range of thresholds or regardless of threshold. Studying cases of intersecting alignment curves is an interesting topic for future work.

---

> > ### Comment · Reviewer_7c5a · 2022-07-29
> > **2nd review**
> >
> > I went through the authors' response, other reviews, and the revised paper. Here is my take on the revised version:
> >
> > -  *causation vs. correlation*: my concerns regarding claiming causality are rectified. The revised paper mainly suggests empirical correlation. While causality would have been stronger with more useful implications, demonstrating empirical correlation of a precisely-defined notion of alignment with the two concepts of generalization and convergence rate would still be interesting by itself.
> > - *Correlation with generalization*: regarding the point of overfitting, the discussion and results in the authors’ answer is both useful and important. It should be noted that an increasing alignment on the training data in tandem with a decreasing performance on the test data suggests an anti-correlation of the considered notion of alignment with generalization in the overfitting phase. This observation would be crucial for the main suggestion of the paper in that it could mean that a general correlation with generalization cannot be suggested. Therefore, corresponding experiments, plots and a proper discussion should be included in the main paper. The suggested correlation with generalization should then be questioned when overfitting happens. Such observation should also be reflected throughout the paper whenever it suggests correlation with generalization. For instance, this can be done by constraining the correlation to the phase before the overfitting happens. Also, I couldn’t find the suggested figures in the appendix. Which part of the appendix demonstrates the alignment on the test set during training? For this, it is better to show the overfitting regime on standard datasets and labels (not random labels) to better reflect the two phases of generalizable learning and overfitting.
> > - *Correlation with convergence rate*: the empirical evidence is similar to the original version which seems insufficient for drawing solid conclusions for convergence rate and alignment.
> > - *Correlation with the positive/negative transfer*: new experiments are added with the Office-31 datasets. The negative transfer is obtained by the extreme case where the domain is changed but also the labels are completely noisy (randomly reassigned). The results indicate that such change in the domain along with the randomization of the labels can cause a negative transfer that also reduces the alignment.
> > - *Regarding the qualitative notion of alignment”: the authors argue that it is intentional for the notion of alignment to remain vague and that the paper only suggests higher/lower alignment when it happens across a wide range of thresholds. Still, in the presentation it is the equation 2 that is called “Alignment” while the text means a different qualitative notion when it refers to the alignment (which is across different thresholds). First, this mismatch should be rectified. Second, it is important to leave no room for subjectivity by either having a precise definition for the used notion of alignment or rewording the observations/discussions to avoid phrases such as “higher/lower alignment” or in some other way.

---

> > > ### Author Response · Authors · 2022-08-03
> > > **Response**
> > >
> > > Thanks for the additional comments.
> > >
> > > Alignment on out-of-sample data: The Figure we mentioned in the previous response is Figure 18 in the appendix. For feature transfer and fine-tuning experiments with large neural networks, alignment plots are computed on a sample from the target task, which is already separate from the sample that the neural networks were trained on.
> > >
> > > Correlation with convergence rate: We have now extended Proposition 1 (Section 6.1) to show the rate of convergence in the loss. Note that unlike most other convergence results, this one is not a bound but an equality. In other words, evolution of the learning curve in this regime is precisely determined by projections of the label vector on singular vectors of the representation and the corresponding singular values. Due to this strong tie in the linear regime and the wealth of evidence in Neural Tangent Kernel regime in the papers we have cited, we find it extremely unlikely that the relationship between alignment and convergence rate in our more general experiments is due to chance. Still, if you believe any other particular experiment is necessary, we will gladly add it to the paper. (The current manuscript with appendix has become 25 pages.)
> > >
> > > Objectivity and the role of threshold: The main message of Section 6.1 was that large alignment at a high threshold means that a large amount of loss will be reduced at a high rate. We have now solidified this intuition into a new proposition (Proposition 2 in the same section). This proposition uses our exact definition of alignment to bound the number of iterations needed to reduce a certain amount of loss. The experiment in Figure 9 is a testament to this proposition. That is, in the case of intersecting alignment curves, whether one representation results in faster convergence than another one depends on how much of the loss we want to reduce or how many iterations the experiment runs for. The reason we report an alignment curve across many thresholds instead of picking a threshold or reducing the curve to a scalar measure in another way is that there is no single way to decide on the number of iterations in a machine learning experiment. Sometimes practitioners stop the experiment when the model's loss appears to have converged (either by visually checking the learning curve or picking an arbitrary tolerance parameter). Other times an early stopping approach based on performance on a validation set is incorporated and so the number of iterations depends on the generalization performance of the model. The number of iterations may also depend on the (possibly non-gradient-based) baselines that the model is compared against if the experimenter wants to give all the models the same amount of computation. All this is hard to formulate in order to choose a certain threshold. We hope that, even if our disagreement on reporting a scalar measure persists after this discussion, the new proposition grounds the definition of alignment in this paper.

---

### Review · Reviewer_HjG1 · 2022-06-24

**Summary Of Contributions:**

The paper introduces the concept of representation alignment. The representation alignment is the degree to which the label vector is aligned (dot product) with largest (above some threshold \tau) singular values of the representation matrix (features for each datapoint from the last hidden layer of the model). The paper plots representation alignment curves, sweeping different values of \tau which provides a measure of how well the representations are aligned with the labels.

The paper then shows that more aligned representations emerge during the training of neural networks under a variety of conditions. RBF networks/dictionary learning produces less well aligned representations. For linear models (one layer trained on top of hidden representations), the paper provides evidence that more aligned representations result in faster convergence. The paper also shows that better alignment between representations and downstream labels results in better transfer.

**Requested Changes:**

* Expand upon the real-data experiments to show that representation alignment holds as a good predictor of transfer performance on more settings than supervised ImageNet->Cifar with a ResNet.

* Provide further evidence or explanation for the cases where representation alignment does not seem to follow the expected pattern (e.g. larger networks appear not always better aligned).

**Strengths And Weaknesses:**

Strengths

* The paper is clearly written, with useful diagrams, and the concept of representation alignment is well explained.

* The paper provides some analytical intuition (proposition 1) as well as empirical evidence for their claims that representation alignment helps training.

* There are a number of carefully controlled synthetic experiments that demonstrate the various points.

* There are some experiments with real data/networks (imagenet->cifar transfer) to test the hypotheses.

Weaknesses

* The real-data experiments are rather limited. The paper studies imagenet->cifar with CNNs (including using custom splits to show that source/task matching is important and increases alignment). However, it is hard to conclude from this limited data that representation alignment is always a good indicator of transfer performance, especially when finetuning, which is more practically useful than training only a linear layer. I think the empirical section would be greatly strengthened if the correlation between representation alignment and better transfer was shown to hold across a variety of different architectures (e.g. CNNs, Transformers) and pre-training strategies (e.g. supervised, self-supervised, auto-encoders).

* The quantification of representation alignment, and showing that this property correlates with improves linear transfer is intuitive. However, various results are presented that are not well explained. For example, in Figure 6 deeper/wider networks do not appear well correlated with improved alignment, although increased depth/width consistently improves performance. It is also observed that random features at initialization are less well aligned than the input features (e.g. Fig 13 (a)). Do these random networks transfer less well than a linear layer on the input? If not, then how does that fit with the rest of the findings?

* The paper claims that representation alignment is the reason for good transfer: "In this work, we developed and investigated alignment as a potential reason for why neural network representations transfer well."  However, I feel this claim is a little strong, the paper shows that aligned representations correlates with good transfer, but does not explain the cause of aligned representations (and hence good transfer).

Typos:

"training with gradient descent on a representations with high alignment" -> “on representations”

“help speed learning” -> “speed up” (or accelerate)

"Training increases alignment in hidden representation" -> "representations"

---

> ### Author Response · Authors · 2022-07-08
> **Response**
>
> Thank you the valuable feedback and positive comments on clarity and presentation.
>
> Real-data experiments: The revision includes the ImageNet->Cifar fine-tuning experiment with a Tokens-to-Token Vision Transformer (which is suitable for ImageNet pre-training). We also added an experiment on negative transfer on six tasks of Office-31 dataset.
>
> Various results are presented that are not well explained: It is possible that MNIST and CT Position dataset are simple enough that there is little or no benefit in increasing depth beyond three layers. The networks in this figure do not have convolutional layers, residual connections or batch normalization. Adding depth to these architectures might have no positive effect or even harm performance [1, 2], which could be why we do not find a consistent increase in final alignment as depth increases. We added this explanation to the revision.
>
> Causal relationship between alignment and transfer: Thank you for mentioning this. We removed this statement in the revision as the experiments are about co-occurrence of alignment and transfer. We leave causal discovery for future work.
>
>
> [1] He, Kaiming, et al. "Deep residual learning for image recognition." CVPR 2016.
> [2] Ioffe, Sergey, and Christian Szegedy. "Batch normalization: Accelerating deep network training by reducing internal covariate shift." ICML 2015.

---

### Review · Reviewer_VSnv · 2022-06-30

**Summary Of Contributions:**

The main contribution of the paper is to develop and investigate alignment as a potential reason for why neural network representations transfer well. Specifically, this paper shows that after training, neural network representations align their top singular vectors to the targets. Further, this paper investigates the above representation alignment phenomenon in a variety of neural network architectures and finds that (1) alignment emerges across a variety of different architectures and optimizers, with more alignment arising from depth (2) alignment increases for layers closer to the output and (3) existing high-performance deep CNNs exhibit high levels of alignment. Finally, this paper highlights why alignment between the top singular vectors and the targets promotes transfer and shows in a classic synthetic transfer problem that representation alignment is the determining factor for positive and negative transfer to similar and dissimilar tasks.


**Broader Impact Concerns:**

This paper has the potential to inspire future works which focus on transfer learning theories and algorithms. Although the work is not very ready for publication, it can provide insights to the community. My only broader impact concern is that the conclusion in Eq(1) may be misleading to the readers.

**Requested Changes:**

Requested changes are all in weaknesses above:
1. The conclusion in Eq(1) should be more rigorious.
2. The picture does not agree with the text in section 4.1.
3. More real-world datasets should be used in empirical observations.
4. How does alignment changes during the fine-tuning processes?
5. How this paper can inspire the design of transfer learning algorithm and narrow the distance from the true solution?

**Strengths And Weaknesses:**

Strengths:
1. The overall writing quality is good. The motivation of the paper is clear and the paper is enjoyable to read.
2. This paper tries to develop and investigate alignment as a potential reason for why neural network representations transfer well. The idea is insightful and convincing.
3. Empirical observations are insightful and have the potential to inspire future works.

Weaknesses:
1. The conclusion in Eq(1) is not very rigorious. If $w_i^V$ and $y_i^U$ are independent. The eq(1) can lead to the conclusion that "it is clear that if $\sigma_i$ is small and the rotated $y_i^U=u_i^{\rm T}y$ is big, then $w_i^V$ has to become very big, which is problematic". However, $U$ and $V$ are all come from the feature matrix $\Phi$ (singular value decomposition). As a result, $U$ and $V$ are not independent, $w_i^V$ and $y_i^U$ are not independent. It's very possible that $\sigma_i$ is small and the rotated $y_i^U=u_i^{\rm T}y$ is big. The present sequence of statements may be misleading. Maybe empirical observations should be put in the front to provide evidences.
2. In section4.1, where is figure 19? Figure 19 in the supp has green and orange lines while in the text there are only yellow and purple lines.
Figure 4 is interesting, the authors should replace Figure 19 with Figure 4.
3. Datasets are not real enough. Most experiments are condducted in simple datasets.
4. This paper measures alignment and convergence rate on Cifar100 using a ResNet101 pre-Trained on ImageNet. The observations only use frozen feature representations. How does alignment changes during the fine-tuning processes, which can be a more important observation for transfer learning setup.
5. How this paper can inspire the design of transfer learning algorithm and narrow the distance from the true solution. Further discussions should be provided.

---

> ### Author Response · Authors · 2022-07-08
> **Response**
>
> Thank you for the feedback and we are glad that you found this work inspiring.
>
> 1. This statement considers the optimal mean squared error solution for a specific representation matrix and label vector. The paragraph discusses how the weight vector has to develop on this particular representation. We edited this sentence to "has to become very big in order to reduce the loss in this task" to make it clear. Please let us know if we misunderstood your point.
>
> 2. Thanks for catching this. It was meant to be Figure 4. It is fixed now.
>
> 3. We added a study of negative transfer on six tasks of Office-31 dataset in the revision. These tasks show different levels of negative transfer which correlates with the reduction in alignment.
>
> 4. The new experiment on negative transfer (Office-31 dataset) considers fine-tuning. We also added an ImageNet->Cifar fine-tuning experiment with a Vision Transformer architecture in the same section.
>
> 5. Our motivation for this work is to pave the way for future algorithmic and theoretical developments in transfer learning. We decided to focus on an empirical study of representation alignment in this paper. Suggesting a new algorithm is left for future work where adequate new results can be provided to support claims about the specific algorithm.

---

### Review · Reviewer_4c69 · 2022-07-07

**Summary Of Contributions:**

The paper suggests that representation "alignment" with target labels plays a role in providing better transfer.
First, the paper defines alignment as the quality where the vector of labels (across samples) is correlated with the top eigenvectors of the representation matrix (feature x samples). Second, they show in experiments on 3 small datasets that this alignment grows during training and monotonically increases with the depth where a layer is located in the network.



**Broader Impact Concerns:**

No specific concerns. This paper is about theoretical analysis of how learned representations fit labels in supervised learning.

**Requested Changes:**

To convince that there is a causal relation between alignment and transfer, the paper should show that changing alignment, rather than changing other properties of the representation leads to better transfer.  TO achieve that goal, the paper should separate alignment from other properties of the representation like information content, manipulate only alignment (while maintaining other properties) and measure the effect on transferability.  I suspect that this is very hard to do.
Instead, the paper can change its main message, and state that it finds that alignment is correlated with transfer

**Strengths And Weaknesses:**

Strengths:
(1) The paper is clearly written and explained. Experiments are well described, ad contain enough details. All quantities are well defined and explained.
(2) The problem of transferability is very important and interesting. In the current context it pertains to generalization to a new classes by retraining top layers,


Weakness:
(1) The most interesting claim of the paper is that there is a causal relation between representation alignment, and better transfer, as the opening states, that alignment "plays an important role" in transfer. The discussion puts forward alignment as a "reason" for why NN transfer well.  The paper does not show that causal relation in a convincing case, but instead shows that alignment is correlated with transfer.  This correlation is demonstrated in one synthetic dataset and also using three ConvNets (VGG, ResNet50, ResNet101) on Imagenet.

(2)  The paper does not do a good job in convincing that alignment, and not other properties of the representation is the key factor in transferability. Maybe there are other properties of a representation that could contribute to transfer that are more fundamental than alignment as defined here?








(2) The empirical demonstration that representations become more aligned to the labels during training is convincing, but the paper does not put that observation in context of standard analysis of deep learning. It does not explain how that observation is different from previous work.  Clearly, when optimizing a representation layer such that it will be linearly related to the label vector through the classification layer w, that optimization tends to map samples such that projecting them using we will order them in a separable way (positive samples on one side, and negatives on another). This means that after mapping samples to the representation, it is expected to see them "sorted" along main axes.

---

> ### Author Response · Authors · 2022-07-08
> **Response**
>
> Thank you for recognizing the importance of the studied problem and for bringing up the issue with claiming a causal relationship. We agree that establishing a causal relationship is hard and we leave it for future work. The sentences that implied a causal relationship are now removed in the revision to ensure consistency between claims and evidence. Please let us know if there is any other statement that should be edited.

---

> > ### Comment · Reviewer_4c69 · 2022-07-23
> > **Is alignment the key factor for transferability?**
> >
> > Thank you for your reply. If the only change in the paper is to remove statements about causality, then the contribution of the paper is to show that the representation alignment is correlated with labels. I was not convinced that this observation would be very useful or provides significant insight.  My main concern remains:  The paper does not convince that alignment, and not other properties of the representation, is the key factor in transferability.

---

> > > ### Author Response · Authors · 2022-07-26
> > > **Significance**
> > >
> > > You are right that we should not have suggested causality, since our results cannot support such a strong claim. This is why we said we would tone down the claims. However, since the theory shows that the convergence rate of gradient descent on squared error is fully dictated by alignment, we do think it is likely that alignment is a factor in convergence rate in more general settings as well, and not that this is simply a coincidence that there is a correlation.
> > >
> > > Even without a causal relationship, correlation by itself can be helpful. If a neural net pre-trained on a related task is expected to have aligned representations for the target task and there is abundant evidence for this then a theoretician can safely include this as an assumption to get more realistic guarantees (especially if lower bounds prohibit deriving such a result without this assumption) or an algorithm designer can design an algorithm that works well under this assumption and not worry if it works poorly in general.
> > >
> > > We believe the purpose of TMLR is to provide work that is of interest to some in the community, without making strong claims about significance. Highlighting that there is this correlation is absolutely of interest to those that can then follow up and potentially use this phenomenon for neural network design and also to themselves investigate potential causal relationships.

---

### Decision · Action_Editors · 2022-08-17

**Recommendation:** Accept as is

**Comment:**

This paper received four detailed reviews with insightful questions and constructive suggestions. After extensive revisions, authors managed to address the majority of comments, in particular through more results. One reviewer still held some reservations towards the paper, such as correlation vs. generalization/negative transfer/convergence rate. AE found these comments very thought-provoking while most of them were responded in an acceptable manner by the authors. The current paper turns out to be a nice study on representation alignment and representation transferability, and provides several useful insights and takeaways for the community of deep learning. It aligns clearly with the TMLR expectations and is recommended for acceptance.